# Anthropogenic aerosol forcing - insights from multiple estimates from aerosol-climate models with reduced complexity

Stephanie Fiedler[1], Stefan Kinne[1], Wan Ting Katty Huang[2], Petri Räisänen[3], Declan O'Donnell[3], Nicolas Bellouin[4], Philip Stier[5], Joonas Merikanto[3], Twan van Noije[6], Risto Makkonen[3,7], and Ulrike Lohmann[2]

[1]Max Planck Institute for Meteorology, Hamburg, Germany
[2]Institute for Atmospheric and Climate Science, ETH Zürich, Zürich, Switzerland
[3]Finnish Meteorological Institute, Helsinki, Finland
[4]Department of Meteorology, University of Reading, Reading, UK
[5]Department of Physics, University of Oxford, Oxford, UK
[6]Royal Netherlands Meteorological Institute, De Bilt, Netherlands
[7]Institute for Atmospheric and Earth System Research / Physics, Faculty of Science, University of Helsinki, Finland

*Correspondence to:* Stephanie Fiedler (stephanie.fiedler@mpimet.mpg.de)

**Abstract.** This study assesses the change in anthropogenic aerosol forcing from the mid-1970s to the mid-2000s. Both decades had similar global mean anthropogenic aerosol optical depths, but substantially different global distributions. For both years, we quantify (i) the forcing spread due to model internal variability and (ii) the forcing spread among models. Our assessment is based on new ensembles of atmosphere-only simulations with five state-of-the-art Earth system models. Four of these models will be used in the sixth coupled model inter-comparison project (CMIP6, Eyring et al., 2016). Here, the complexity of the anthropogenic aerosol has been reduced in the participating models. In all our simulations, we prescribe the same patterns of the anthropogenic aerosol optical properties and associated effects on the cloud droplet number concentration. We calculate the instantaneous radiative forcing (RF) and the effective radiative forcing (ERF). Their difference defines the net contribution from rapid adjustments. Our simulations show a model spread in ERF from -0.4 to -0.9 Wm$^{-2}$. The standard deviation in annual ERF is 0.3 Wm$^{-2}$, based on 180 individual estimates from each participating model. This result implies that identifying the model spread in ERF due to systematic differences requires averaging over a sufficiently large number of years. Moreover, we find almost identical ERFs for the mid-1970s and mid-2000s for individual models, although there are major model differences in natural aerosols and clouds. The model-ensemble mean ERF is -0.54 Wm$^{-2}$ for the pre-industrial era to mid-1970s and -0.59 Wm$^{-2}$ for the pre-industrial era to mid-2000s. Our result suggests that comparing ERF changes between two observable periods rather than absolute magnitudes relative to a poorly constrained pre-industrial state might provide a better test for a model's ability to represent transient climate changes.

## 1 Introduction

Despite decades of research on the radiative forcing of anthropogenic aerosol, quantifying the present-day magnitude and reconstructing the historical change of the forcing remains challenging. Figure 1 shows the anthropogenic aerosol optical

depth for the mid-1970s and mid-2000s that we use in this study (Fiedler et al., 2017; Stevens et al., 2017). The anthropogenic aerosol pollution in the mid-1970s was larger in Europe and North America than in East Asia, whereas the opposite is the case in the mid-2000s. In addition to these regional changes in aerosol pollution, differences in the surface albedo, insolation, and cloud regimes between the aerosol transport regions of the Pacific and continental Europe may result in temporal changes in

the global effective radiative forcing (ERF). Based on a single state-of-the-art climate model, the long-term and global ERF does not change despite the substantial spatial changes in anthropogenic aerosol optical depth ($\tau_a$) between the mid-1970s and mid-2000s (Fiedler et al., 2017). Internal model variability, however, strongly affects annual estimates of the global mean effective radiative forcing.

In light of model uncertainties (e.g., Kinne et al., 2006; Quaas et al., 2009; Lohmann and Ferrachat, 2010; Lacagnina et al.,

2015; Koffi et al., 2016), the use of a single model does not necessarily represent the full spectrum of possible anthropogenic aerosol forcings. In the present study, we therefore revisit the question of Fiedler et al. (2017): "Does the substantial spatial change of the anthropogenic aerosol between the mid-1970s and mid-2000s affect the global magnitude of ERF?", based on ensembles of simulations from five global aerosol-climate models, all using identical anthropogenic aerosol perturbations of reduced complexity. In this context, we additionally ask: "What is the relative contribution of internal model variability to the

ERF spread?", and document the model diversity for the pre-industrial aerosol as well as cloud characteristics and the surface albedo that are relevant for the ERF of anthropogenic aerosol. Such model differences have previously been identified for other climate models (e.g., Stier et al., 2007; Nam et al., 2012; Fiedler et al., 2016; Crueger et al., 2018).

Previously a reduction of the model complexity has been accomplished by prescribing idealized aerosol radiative properties, e.g., within the framework of Aerosol Comparisons between Observations and Models (AeroCom, e.g., Randles et al., 2013;

Stier et al., 2013). Here, we prescribe observationally constrained optical properties of anthropogenic aerosol and an associated effect on the cloud droplet number concentration with the simple plumes parameterization (MACv2-SP, Fiedler et al., 2017; Stevens et al., 2017), but keep the full model diversity in all other aspects. The approach eliminates uncertainties in process modelling of anthropogenic aerosol such that our study represents uncertainties associated with other processes influencing the radiative forcing. In other words, by using MACv2-SP in the participating models, the model inter-comparison allows us

to investigate those sources of uncertainty that remain if we pretend to know the spatial distribution of anthropogenic aerosol. This work can be seen as a pilot study for the "Radiative Forcing Model Inter-comparison Project" (RFMIP, Pincus et al., 2016), endorsed by CMIP6 (Eyring et al., 2016), using the same experiment setup with MACv2-SP.

Throughout our model inter-comparison, we consider the effect of model-internal variability on estimates of ERF. We do so by producing equally-sized ensembles of simulations for all participating models. Model-internal variability in this context

is defined as the year-to-year changes in model parameters associated with inter-annual variations of the meteorological state. The results of the climate models are compared with satellite data and a stand-alone radiative transfer model. The following section introduces the models and the experiment strategy in more detail, followed by our discussion of the results in Section 3 and conclusions in Section 4.

## 2 Method

### 2.1 Participating Models

This work uses five Earth system models and one stand-alone radiative transfer code. The participating climate models, which are run here in an atmosphere-only mode, are the atmosphere component ECHAM6.3 of the Earth system model MPI-ESM1.2 (Mauritsen et al., 2019) of the Max-Planck Institute for Meteorology (MPI-M), ECHAM6.3-HAM2.3 from the ETH Zürich (Tegen et al., 2018; Neubauer et al., submitted to ACPD), EC-Earth (e.g., Hazeleger et al., 2010; Döscher et al., in prep.) run at the Royal Netherlands Meteorological Institute, NorESM (Bentsen et al., 2013; Iversen et al., 2013; Kirkevåg et al., 2013) run at the Finnish Meteorological Institute, and HadGEM3 (Walters et al., 2017) developed at the UK Met Office. All models except ECHAM6.3 can treat aerosols and their interaction with meteorological processes with complex process-based parameterisation schemes linking aerosols to radiation and clouds. In this study, all physics packages except the parameterization of anthropogenic aerosols are model-dependent, e.g., the treatment of the pre-industrial aerosols and clouds differ. Appendix A summarizes differences in radiation, cloud, and aerosol physics packages of the participating models.

In the present study, we prescribe the distributions of anthropogenic aerosols in all models following the MACv2-SP approach (Fiedler et al., 2017; Stevens et al., 2017). MACv2-SP mimics the spatio-temporal distribution and wavelength dependence of the optical properties of anthropogenic aerosols as well as a change in the cloud droplet number concentration ($N$) to induce radiative effects associated with the physical processes of aerosol-radiation interactions ($F_{ari}$) and aerosol-cloud interactions ($F_{aci}$) in a consistent manner. To do so, MACv2-SP uses analytical functions for approximating the monthly distribution of the present-day anthropogenic aerosol optical depth and the vertical profile of the aerosol extinction from the updated MPI-M aerosol climatology (MACv2, Kinne et al., 2013, Kinne et al, in prep.). Figure 1 shows the annual mean patterns of the anthropogenic aerosol optical depth ($\tau_a$), and the fractional increase in the cloud droplet number concentration ($\eta_N$) relative to the pre-industrial level of 1850 from MACv2-SP. By design, MACv2-SP does not simulate sub-monthly variability in anthropogenic aerosol. Absorption of anthropogenic aerosol is prescribed with a mid-visible single scattering albedo of 0.93 for industrial plumes and 0.87 for plumes with seasonally active biomass burning. The anthropogenic aerosols are assumed to be small in size with an Angstrom parameter of 2 and an asymmetry parameter of 0.63. Here, we use MACv2-SP with the CMIP6 reconstructed changes of anthropogenic aerosol emissions, identical to the one used by Fiedler et al. (2017). Stevens et al. (2017) describe the technical details of MACv2-SP.

The use of the optical properties from MACv2-SP yields a consistent description of $F_{ari}$, including both direct radiative and semi-direct effects, across the models. All models account for the first indirect or Twomey effect by multiplying their cloud droplet number concentrations, calculated for pre-industrial aerosol conditions, by $\eta_N$ prior to the radiative transfer calculation. Since $\eta_N$ is larger than one in the presence of anthropogenic aerosols, the effective radius of cloud droplets is reduced, which enhances the cloud reflectivity of shortwave radiation. Note that $\eta_N$ is only available for regions with $\tau_a$>0 (see Fig. 1). In addition, the EC-Earth model also includes a second indirect or cloud lifetime effect by using the modified cloud droplet number concentrations in the cloud microphysics scheme (Döscher et al., in prep.).

We do not prescribe the same natural aerosol nor interfere with any other model components than prescribing the optical properties of anthropogenic aerosols and $\eta_N$. For instance, the pre-industrial aerosol optical depth ($\tau_p$) depends on the model (Figures 2 and 3), which only affects $F_{ari}$ and not $F_{aci}$ as the prescription of $\eta_N$ is identical in the participating models. Regional differences in $\tau_p$ occur primarily over oceans and deserts, where observations are typically sparse. It is noteworthy

that ECHAM-HAM runs with interactive parameterisations for dust and sea-salt aerosol resulting in different spatio-temporal variability in $\tau_p$ (Figure 3), while in ECHAM the monthly climatology from MACv1 is prescribed. In the interactive parameterisations, the natural aerosol emissions, transport and deposition rely on meteorological processes that are difficult to represent in coarse-resolution climate models, e.g., desert-dust emissions strongly depend on the model representation of near-surface winds (e.g., Fiedler et al., 2016) such that constraining the desert-dust burden remains challenging in aerosol modelling (e.g.,

Räisänen et al., 2013; Evan et al., 2014; Huneeus et al., 2016). The aerosol-climate models also contain some anthropogenic aerosol in $\tau_p$, but the majority of the pre-industrial aerosol optical depth is of natural origin. For instance, the 1850's global-mean $\tau_p$ in NorESM is 0.096, to which anthropogenic fossil-fuel aerosols contribute 0.002. For comparison, the here prescribed global mean $\tau_a$ is 0.029 for 2005.

In addition to the complex climate models listed above, we use the offline radiative-transfer model of Kinne et al. (2013)

for an assessment of the instantaneous radiative forcing. This model has eight solar and twelve infrared bands, and reads monthly maps of the atmospheric and surface properties. These are for instance monthly means for the cloud properties from ISCCP and the surface albedo from the satellite product MODIS-SSM/I (Kinne et al., 2013). The radiative transfer calculation considers nine different sun elevations and eight randomly chosen combinations of cloud heights and overlap. The aerosol column properties at 550 nm are defined by the MPI-M Aerosol Climatology (MAC). The aerosol vertical distribution and the

fine-mode anthropogenic fraction of aerosol optical depth for the mid-2000s are derived from global models participating in AeroCom (e.g., Myhre et al., 2013). We calculate the radiation transfer with both MAC version one (MACv1, Kinne et al., 2013) and two (MACv2, Kinne, in review). The latter considers more recent observational data, e.g., from the Maritime Aerosol Network (MAN, Smirnov et al., 2009), and a smaller anthropogenic aerosol fraction. MACv2 is also based on more recent emission data relative to 1850 (Lamarque et al., 2010), while MACv1 used emission data relative to 1750 (Dentener et al.,

2006). The two climatologies therefore make different assumptions on the pre-industrial background, shown in Figure 3. The temporal scaling of anthropogenic aerosol optical depth in MACv1 and MACv2 is from the same transient ECHAM simulation (Stier et al., 2006). The parameterization form of the Twomey effect for MACv1 and MACv2 are here identical to MACv2-SP, but the assumptions for $\tau_p$ and $\tau_a$ differ.

## 2.2 Experiment strategy

All climate model simulations are carried out with the atmosphere-only configurations using prescribed monthly mean sea-surface temperatures and sea ice. Table 1 summarises the major characteristics of the model simulations. The modelling groups were free to set up all model components other than MACv2-SP, and choose their own boundary and initialization data. Specifically, the modelling groups use their own representation of pre-industrial aerosol for 1850 such that the present work includes

both models with prescribed monthly climatologies and interactive parameterisation schemes for natural aerosol species (Appendix A). Moreover, the physical parameterisations of radiation and clouds are different across the models (Appendix A).

Motivated by the effect of natural variability on ERF estimates in ECHAM (Fiedler et al., 2017), each model was run to produce a number of simulation ensembles: a reference ensemble consisting of six simulations with only pre-industrial aerosols representative for 1850, and two additional ensembles consisting of three simulations each with aerosols representative for 1975 and 2005, respectively. For each model, we perform a total of twelve experiments for the years $2000-2010$ inclusive. These are six experiments with $\tau_p$ for the year 1850, three experiments with $\tau_p$ and anthropogenic aerosol from MACv2-SP for the year 1975, and three experiments with $\tau_p$ and anthropogenic aerosol from MACv2-SP for the year 2005. The six pre-industrial simulations serve as the reference for the experiments with anthropogenic aerosol and therefore efficiently increase the number of forcing estimates for anthropogenic aerosol. The first year of each run is considered as a spin-up period and is excluded from the analysis. A ten-year period was chosen to account for variability in the boundary conditions.

The instantaneous radiative forcing (RF) of anthropogenic aerosols in clear and all sky is estimated from double radiation calls in the models having this functionality, namely ECHAM, ECHAM-HAM and NorESM. Aerosol radiative effects predominantly occur for shortwave radiation. We therefore calculate the atmospheric transfer of shortwave radiation once with and once without the contribution from anthropogenic aerosols to the aerosol optical properties and their effect on the cloud droplet number concentration. For each model, this gives us in total 30 annual estimates of RF for each of the two $\tau_a$ patterns shown in Figure 1, which is sufficient to estimate the mean RF and can be directly compared to the offline radiation-transfer calculations. We calculate RF at the top of the atmosphere (TOA) and at the surface (SFC) and list the global means in Table 2.

The ERF is calculated as the difference in the shortwave radiative flux at the top of the atmosphere between the simulations with and without anthropogenic aerosols. For illustrating the effect of year-to-year variability, we calculate annual ERF estimates for each of the ten simulation years. Combining the six pre-industrial experiments with each of the three experiments with additional anthropogenic aerosol thus yields 6x3 annual ERF estimates for each year of the simulation, i.e., 180 annual estimates per model and $\tau_a$ pattern in total. We calculate the standard deviation from these 180 annual ERF values and use it as a measure of the natural variability in ERF internal to the models. The means of these 180 values are used for identifying systematic model differences in ERF. It was shown in an earlier study using ECHAM (Fiedler et al., 2017) that the combination of ensemble size and simulation length adopted here is sufficient for precisely estimating the ERF of a model. For comparison, the RFMIP protocol recommends a thirty-year average for diagnosing the ERF of a model (Pincus et al., 2016). Finally, we calculate the net contribution of rapid adjustments (ADJ) to ERF by subtracting RF from ERF for each model. Our rapid adjustments are associated with atmospheric temperature changes, i.e., semi-direct effects, except for EC-Earth accounting also for adjustments in cloud microphysics. A discussion of the rapid adjustments and the choice for the Twomey effect in ECHAM is given by Fiedler et al. (2017).

## 3 Results

### 3.1 Spread in present-day ERF

We characterise the spread in the shortwave effective radiative forcing (ERF) at the top of the atmosphere in our model ensemble for the present-day (mid-2000s). For doing so, we first calculate the multi-model mean as a reference value. The all-sky top-of-atmosphere ERF for the entire multi-model, multi-member ensemble is $-0.59$ Wm$^{-2}$ with an interannual standard deviation of 0.3 Wm$^{-2}$, corresponding to a relative variability of roughly 50%. The interannual variability in ERF is illustrated by Gaussian distributions fitted to the frequency histogram in Figure 4a. The entire range in annual ERFs from the models including interannual variability is $-1.5$ Wm$^{-2}$ to $+0.5$ Wm$^{-2}$.

The all-sky ERFs from the models are $10-50\%$ less negative than the clear-sky ERF in all models, except in EC-Earth, because clouds mask the ERF of low-level anthropogenic aerosol (Table 2). That masking by clouds is most pronounced in HadGEM3. In EC-Earth, the all-sky ERF is more negative than in clear-sky because EC-Earth includes cloud lifetime effects of anthropogenic aerosols, thus simulating a stronger $F_{aci}$ than all other participating models. The long-term averaged ERFs of ECHAM and ECHAM-HAM are similar, despite ECHAM using a prescribed climatology of $\tau_p$ and ECHAM-HAM simulating $\tau_p$ interactively (Section 2.1). This similarity suggests that the sub-monthly variability of natural aerosol does not substantially affect the mean ERF of anthropogenic aerosol, as long as $F_{aci}$ is treated consistently in the two models. Using different parameterizations for $F_{aci}$ can change this result because of non-linear processes. The magnitude of $F_{aci}$, however, remains uncertain (Bellouin et al., in prep.). One contributing uncertainty is the poor quantitative understanding of the pre-industrial aerosols (e.g., Carslaw et al., 2013).

The multi-model spread in the ensemble mean all-sky ERF of individual models is rather small, with a range of $-0.40$ Wm$^{-2}$ to $-0.9$ Wm$^{-2}$, compared to the internal variability of the entire multi-model ensemble (Fig. 4a). This multi-model spread corresponds to a range of deviations from the multi-model mean of just $-0.31$ Wm$^{-2}$ to $+0.19$ Wm$^{-2}$ and is even smaller when the ERF of EC-Earth, which includes cloud-lifetime effects, is excluded. One could expect less model diversity in all-sky ERF from our study than from previous inter-comparison projects (e.g., Myhre et al., 2013; Shindell et al., 2013), because we prescribe the same aerosol optical properties and the associated change in cloud droplet numbers. However, our model diversity in clear-sky ERF is smaller than for our all-sky ERF (Table 2). This points to the influence of model differences in representing clouds (Appendix B) on the all-sky ERF. Our results therefore indicate that model differences in meteorological parameters contribute to the model diversity in all-sky ERF. This is also the case for the ERF uncertainty in a complex aerosol-climate model (Regayre et al., 2018).

The large interannual variability implies that it is essential to estimate ERF of individual models from a sufficiently large number of simulated years to quantify model differences in ERF. Otherwise the modelled ERF estimates may not be representative of the long-term average. This could be done either from sufficiently long simulations with annually repeating aerosol or a sufficiently large ensemble of simulations with transient changes. Given the similar year-to-year variability in ERF in the models, the confidence estimates from ECHAM (Fiedler et al., 2017) are a reasonable approximation for the whole ensemble of models in the present study.

## 3.2 Regional contributions to ERF

The distributions of ERF for 2005 are shown as ensemble averages in Figure 5 and for each model in Figure 6. East Asia is the largest contributor to globally-averaged ERF, as expected from the regional maximum in $\tau_a$ prescribed there (Figure 5b). The mean pattern of regional contributions to ERF is in general similar in the models but differences in its magnitude and detectability appear in some regions. For example, the contributions to the global ERF modelled over central Africa range from positive to negative, averaging to a small value in this region (Fig. 5).

Another interesting example for where regional contributions to globally-averaged ERF differ is the North Atlantic. In this region, the variability of the multi-model ensemble is relatively large, $3-6\,\mathrm{Wm^{-2}}$ (Figure 4b), but the small multi-model mean radiative effects are nevertheless detectable (Figure 5), although ECHAM and HadGEM by themselves have regional signals over the North Atlantic that are not statistically significant.

Taken together, the size of year-to-year variability and regional model differences in contributions to the global ERF imply that an ensemble of simulations with more than one model, as done here, is needed for constraining the radiative effect of anthropogenic aerosol regionally. The spread in modelled regional contributions to ERF is typically smaller than the differences associated with natural variability in the model ensemble (Figure 4b−c). Irrespectively whether we compute the regional standard deviations for the aerosol pattern of the mid-1970s or the mid-2000s, the pattern and strength of the regional natural variability in contributions to ERF is robust (not shown). In regions where the anthropogenic aerosol burden was relatively large in 2005, like East Asia, the models disagree on the magnitude of the regional contributions to ERF (Figure 4c), which means that even for a relatively large anthropogenic aerosol optical depth, natural variability of the atmosphere remains a hurdle against constraining the regional radiative effect.

## 3.3 Contributions from RF and adjustments

The modelled ERF is decomposed into the contributions of rapid adjustments and RF by diagnosing the latter from double calls to the radiation scheme in the models with this functionality (Figure 5). The RF is considerably less variable from year to year than ERF. Moreover, RF clearly dominates the ERF magnitude in all models that use $\eta_N$ in the radiation transfer calculation (Table 2). Remember that these models consider $F_{\mathrm{aci}}$ from the Twomey effect only. The net contribution of rapid adjustments to the global mean ERF ranges from $0.03\,\mathrm{Wm^{-2}}$ in NorESM to $0.2\,\mathrm{Wm^{-2}}$ in ECHAM-HAM at TOA, and acts to weaken the forcing magnitude. The positive net contribution from adjustments is consistent with buffering of perturbations by atmospheric processes.

We compare the climate model estimates of RF with the results of the offline radiation-transfer calculations described in Section 2.1. The offline estimates of the all-sky RF with MACv2-SP (Offline-v1-SP and Offline-v2-SP) are in close agreement with the RF of the climate models that represent $F_{\mathrm{aci}}$ in form of the Twomey effect. This agreement is remarkable since the aerosol-climate models and the offline model differ in many aspects, including again the representation of clouds (see Appendix B).

### 3.4 Uncertainties in RF

The offline radiation-transfer model is used to assess the role of uncertainty in $\tau_p$ and $\tau_a$ in total RF uncertainty. The aerosol classification of MACv2 (Offline-v2) is used as an alternative representation to MACv1 (Offline-v1). MACv2 classifies more ambiguous cases of fine-mode aerosol as anthropogenic than MACv2-SP. These cases primarily occur in remote uninhabited regions such as the Southern Ocean and the Saharan desert. These regions are poorly captured by the ground-based observation network so there the MACv2 product primarily uses global model results for separating anthropogenic from natural aerosols. Classifying additional fine-mode aerosol as anthropogenic increases the all-sky RF to $-1.1$ Wm$^{-2}$, which primarily arises due to stronger $F_{\mathrm{aci}}$ in MACv2. Ambiguous aerosol classifications, which occur especially in regions with a generally low aerosol burden, and a poor observational coverage are therefore causes of uncertainty in present-day RF, with the RF getting more negative with increasing $\tau_a$.

An even more negative RF is obtained from the offline model, namely an all-sky RF of $-1.4$ Wm$^{-2}$, when both a larger anthropogenic fraction and the lower background burden of 1750 from MACv1 (Offline-v1) is used. Note that the clear-sky RFs from the offline estimates and the climate models are in good agreement, such that most of the uncertainty stems from the uncertain magnitude of $F_{\mathrm{aci}}$. This underlines again the importance of the aerosol background for quantifying the cloudy-sky contribution to all-sky RF in agreement with previous studies (Carslaw et al., 2013; Fiedler et al., 2017). Quantitative changes in natural aerosol burden between the pre-industrial and present-day remain poorly constrained. Since the aerosol of 1750 or 1850 has not been observed, using the present-day natural aerosol as a background could yield a better comparability of observational and model estimates in future inter-comparison studies. By prescribing both the same natural and anthropogenic aerosol across different models, differences in the radiative effects of the aerosol can be attributed to model errors in representing meteorological processes and radiative transfer.

### 3.5 Impact of spatial change of pollution

Although the global mean $\tau_a$ is similar for 1975 and 2005, the anthropogenic pollution covers very different regions, with the largest maxima in Europe and the U.S. during the mid-1970s and in East Asia during the mid-2000s. The regional differences in clouds, insolation and surface albedo can contribute to changes in radiative effects that can result in a different global ERF. For instance, Figures A1-A3 show the spatial patterns of cloud properties and the surface albedo illustrating both the regional differences and the model diversity for their representation (see Appendix B). The different spatial distributions of $\tau_a$ clearly change the pattern of the radiative forcing (Figure 7). As expected, the maxima in regional contributions to RF and ERF occur over Europe and the U.S. in the mid-1970s and over East Asia for the mid-2000s.

Despite those regional differences in radiative effects and the inter-model spread in ensemble-averaged global mean RF and ERF, the spatial pattern of $\tau_a$ has little impact on the global mean RF and ERF in each of the participating models. The model ensemble mean changes from $-0.54$ Wm$^{-2}$ for the mid-1970s to $-0.59$ Wm$^{-2}$ for the mid-2000s. The mean monthly contributions to RF are also similar for both $\tau_a$ patterns, irrespectively which model we choose (not shown).

The ensemble-averaged change in ERF is small relative to the natural interannual variability in modelled ERFs (Figure 8). Indeed, contrasting one-year estimates from the two $\tau_a$ patterns results in a large spread in ERF changes ranging from decreases to increases in ERF with $\tau_a$ patterns (Figure 8c−d). This result is in agreement with previous findings based on ECHAM only (Fiedler et al., 2017). The result underlines again the importance of using a large number of simulated years for determining changes in ERF from free-running climate models. Moreover, it provides evidence that the global mean ERF does not strongly depend on the regional distribution of anthropogenic aerosol in the northern hemisphere.

The cloudy- and clear-sky contributions to the all-sky efficiency of the ERF, in other words the ratio of ERF to $\tau_a$, helps to better understand why the two $\tau_a$ patterns yield similar ERFs. All-sky efficiency is the sum of contributions from cloudy and clear-sky conditions:

$$\frac{\text{ERF}_\text{all}}{\tau_a} = f\frac{\text{ERF}_\text{cloudy}}{\tau_a} + (1-f)\frac{\text{ERF}_\text{clear}}{\tau_a}, \tag{1}$$

where $f$ is the total cloud fraction, and $\text{ERF}_\text{cloudy}$ and $\text{ERF}_\text{clear}$ the ERF in cloudy and clear-sky conditions, respectively.

Figure 9 shows the regional distribution from the multi-model ensemble average of the terms of Equation 1. The all-sky efficiency often increases with increasing distance to major pollution sources because of the decreasing background aerosol, up to $-100$ Wm$^{-2}$ per unit $\tau_a$. These all-sky efficiencies are primarily explained by the cloudy-sky contributions. Large efficiencies occur typically in remote areas including some regions at the edges of $\tau_a$ plumes (Fig. 9). No clear saturation of $F_{aci}$ is evident at all edges of the $\tau_a$ plumes. Also the spatial distribution of both the all- and cloudy-sky efficiency is rather inhomogeneous. The inhomogeneity contrasts with the clear-sky efficiency, which has much smaller spatial variability.

Averaged globally, all-sky forcing efficiencies for the two aerosol patterns are similar at $-26$ Wm$^{-2}$ per unit $\tau_a$. The regional all-sky ERF efficiencies, however, change between the mid-1970s and mid-2000s (Fig. 9). This change is almost exclusively explained by the cloudy-sky contribution to the ERF efficiency, reflecting the regional change in $\eta_N$ from the mid-1970s to mid-2000s. The strong change in the cloudy-sky contribution is in strong contrast to the relatively minor changes in the clear-sky contributions. Differences in regional efficiencies of anthropogenic aerosol effects on clouds thus balance in the global mean and result in similar global ERFs for the mid-1970s and mid-2000s.

Of all models, NorESM and EC-Earth have the strongest ERF efficiencies around $-30$ and $-40$ Wm$^{-2}$ per unit $\tau_a$, respectively, i.e., the same aerosol perturbation in these two models is much more efficient in inducing effective radiative effects than in the other models, consistent with the more negative ERFs (Fig. 8). In EC-Earth, the more negative ERF arises from also perturbing the cloud microphysics with $\eta_N$. In NorESM, the more negative ERF arises from a strong negative RF and a small net contribution from adjustments.

## 4 Conclusions

We assess the radiative effects of anthropogenic aerosol in ensembles of simulations from five state-of-the-art aerosol climate models, prescribing identical anthropogenic aerosol properties of reduced complexity. Each of the participating models uses annually repeating patterns of anthropogenic aerosol for obtaining 180 years of radiative forcing estimates. The multi-model

multi-ensemble present-day all-sky short-wave effective radiative forcing (ERF) at the top of atmosphere is -0.59 $\text{Wm}^{-2}$. The year-to-year standard deviations of around 0.3 $\text{Wm}^{-2}$ in the models imply a typical year-to-year variability of 50%, reflecting a strong contribution of model internal variability to ERF. We therefore recommend caution for the use of ERF estimates based on single years, as in the standard AeroCom protocol with varying reference years. These are likely affected by model-internal
variability such that an apparent ERF spread is not associated with systematic model differences alone. Indeed such studies have shown a substantial spread in ERF estimates (e.g., Shindell et al., 2013), comparable to the magnitude of the model internal variability quantified in the present work.

We further recommend that model-based assessments of ERF in the future ensure to eliminate the effects of internal variability, either by averaging over longer time periods from single transient climate simulations or from averaging across several
ensemble members for shorter time periods. For instance, the protocol of RFMIP requests thirty-year averages for estimating the present-day ERF and three-member ensembles with ten-year averages for diagnosing decadal changes in ERF (Pincus et al., 2016). The precision of the estimate can be tested by using confidence estimates (e.g., Fiedler et al., 2017). Note that natural variability is equally an issue in observations. Ensembles of simulations should therefore be used for constraining ERF with the historical record of observations. The interannual variability in ERF, and hence the number of years needed to estimate ERF,
could be different in nudged model simulations (Zhang et al., 2014). However, nudging a model simulation with re-analysis data can change the climatology and interfere with the rapid adjustments. The resulting ERFs from a nudged simulation are therefore likely different compared with free-running model simulations. The interference of nudging with adjustments deserves closer attention in future research.

In our study, we obtain an ERF spread of -0.9 to -0.4 $\text{Wm}^{-2}$ associated with systematic model differences (Fig. 10). This
estimate is not affected by model-internal variability, is based on identical anthropogenic aerosol optical properties and makes use of a consistent perturbation of the cloud droplet number concentrations associated with anthropogenic aerosol. The model with the most negative ERF accounts also for changes in cloud microphysics associated with anthropogenic aerosol, whereas the other participating models account for the Twomey effect only. Based on our model spread, we conclude that models with a strongly negative ERF have particularly strong contributions from anthropogenic aerosol effects on clouds.

Our results highlight that the participating models consistently show little change in the global ERF of anthropogenic aerosol between the mid-1970s and mid-2000s, despite the substantially different location of anthropogenic pollution maxima and the model diversity in their ERF magnitude relative to the pre-industrial. Model internal variability, however, produces ERF changes of different signs and magnitude between the two periods. This result gives further evidence that model-internal variability has not been sufficiently considered in past model studies estimating the ERF difference associated with the mid-
1970s to mid-2000s change in anthropogenic aerosol, as previously suggested based on ECHAM alone (Fiedler et al., 2017). The small change in global ERF stems from similar global forcing efficiencies of anthropogenic aerosol in the two periods. These are primarily explained by globally compensating differences in regional cloudy-sky contributions to the ERF efficiency. Assuming stronger aerosol-cloud interactions can cause a larger change in ERF from the mid-1970s to mid-2000s, based on simulations with ECHAM (Fiedler et al., 2017). The forcing from aerosol-cloud interaction is a subject of ongoing discussion
and research (Bellouin et al., in prep.). Given our multi-model spread in absolute ERF relative to the pre-industrial, inter-

comparing the relative ERF changes between observable periods might provide a better test for a model to represent transient climate changes. Our future work will focus on inter-comparing modelled ERF changes associated with other aerosol patterns. One such endeavour is the usage of MACv2-SP in model simulations in the framework of CMIP6 (e.g., Pincus et al., 2016; Fiedler et al., 2018).

*Data availability.* The model data of this study will be available on the AeroCom community's data server. Additionally, the model data is archived by the Max Planck Institute for Meteorology and can be made accessible by contacting publications@mpimet.mpg.de.

## Appendix A: Model physics packages

ECHAM6.3 is the latest version of the atmosphere component of the Earth system model MPI-ESM1.2 of MPI-M, which participates in CMIP6 (Mauritsen et al., 2019). ECHAM6.3 is a global hydrostatic model and includes parameterisations of
sub-grid scale physical processes. The atmospheric radiative transfer is parameterised with the PSrad scheme using the Rapid Radiative Transfer Model for general circulation models (RRTMG, Pincus and Stevens, 2013). Surface properties, trace gas concentrations, and natural aerosols are prescribed by climatological data sets. A major change in MPI-ESM1.2 (Mauritsen et al., 2019) compared to previous model versions is the implementation of MACv2-SP (Fiedler et al., 2017; Stevens et al., 2017).

The global aerosol-climate model ECHAM6.3-HAM2.3 is an updated version of the model described by Tegen et al. (2018) and Neubauer et al. (submitted to ACPD). Revisions made in ECHAM6.3-HAM2.3 relate to the atmospheric model and the description of sea-salt emissions, which have been made dependent on the sea-surface temperature. The model uses ECHAM6.3, but is coupled to the aerosol module HAM (Stier et al., 2005; Zhang et al., 2012). An important difference in the atmospheric components is that ECHAM6.3 uses a single-moment cloud microphysics parameterisation, while ECHAM6.3-HAM2.3 has
a two-moment stratiform cloud scheme (Lohmann and Hoose, 2009) for representing the activation of aerosols as cloud condensation nuclei and ice nuclei in mixed phase clouds. Emission schemes for sea salt (Long et al., 2011; Sofiev et al., 2011), desert dust (Tegen et al., 2002; Cheng et al., 2008), and oceanic dimethyl sulphide (DMS, Nightingale et al., 2000) are run online. Emission of all other aerosol species are prescribed from external input files (Stier et al., 2005; Lamarque et al., 2010). In the configuration used in this study, we prescribe the pre-industrial background of aerosol components from HAM that are
not simulated online. These, in combination with the online-computed natural aerosol emissions, are the only aerosols seen by the two-moment cloud microphysics parameterisation in this study.

EC-Earth (Hazeleger et al., 2010; Döscher et al., in prep.) uses the Integrated Forecasting System (IFS) of the European Centre for Medium-range Weather Forecasts (ECMWF) as its atmosphere component. The latest generation of the model, EC-Earth3, is based on the ECMWF seasonal prediction system 4 with IFS cycle 36r4. The radiation scheme is based on
the Rapid Radiative Transfer Model (Mlawer and Clough, 1998; Iacono et al., 2008) with 14 bands in the shortwave and 16 bands in the longwave spectrum, and uses the Monte-Carlo Independent Column Approximation (McICA) approach (Pincus

and Morcrette, 2003). Many new features have been added to IFS by the EC-Earth consortium. The pre-industrial tropospheric aerosol climatology that is used in combination with MACv2-SP, has been constructed from a simulation with the TM5 aerosol-chemistry model (Huijnen et al., 2010; van Noije et al., 2014), driven by meteorological data from ERA-Interim for the early 1980s. This simulation used CMIP6 emissions of aerosol and precursor gases for 1850, and provides the monthly mean aerosol mass and number concentrations as well as the aerosol optical properties. Stratospheric aerosols are prescribed using the CMIP6 data set of radiative properties. Aerosol-cloud interactions are implemented only for liquid phase, stratiform clouds. The cloud droplet number concentration, $N$, is diagnosed using the activation scheme of Abdul-Razzak and Ghan (2000) and is here modified by $\eta_N$ from MACv2-SP. Cloud microphysics depends on $N$ through autoconversion of cloud droplets to rain. The model used in this study is EC-Earth version 3.2.3. It is close to the CMIP6 version described by Döscher et al. (in prep.), but does not include the latest revisions that were introduced after the simulations for this study were started. Most relevant for this study is that in the CMIP6 version the pre-industrial aerosol climatology has been updated, by changing the parameterization of the production of sea spray in the underlying TM5 model. Specifically, the whitecap coverage has been made dependent on sea-surface temperature, while its power-law dependence on the 10m-wind speed has been changed from the W10 expression proposed by Salisbury et al. (2013) to the expression proposed by Monahan and Muircheartaigh (1980). The main effect of this revision is an increase in aerosol and cloud droplet number concentrations over the Southern Ocean.

Simulations with the Hadley Centre Global Environment Model (HadGEM) use a modified version of the HadGEM3 Global Atmosphere 7.0 climate model configuration (Walters et al., 2017). HadGEM3 normally uses the Global Model of Aerosol Processes (GLOMAP, Mann et al., 2010) to simulate aerosol mass and number, and interactions of aerosols with radiation, clouds and atmospheric chemistry. That scheme is here replaced with prescriptions of the three-dimensional distributions of aerosol extinction and absorption coefficients averaged over HadGEM's 6 shortwave and 9 longwave wavebands, waveband-averaged aerosol asymmetry, and $N$. Those prescriptions are made of three components. First, pre-industrial aerosol and $N$ distributions are taken from a HadGEM3/GLOMAP simulation using CMIP6 emission datasets for the year 1850. Second, stratospheric aerosols are taken from the CMIP6 climatologies for the year 1850. Prescribed $N$ are used in the calculation of cloud albedo (Jones et al., 2001) and autoconversion rates (Khairoutdinov and Kogan, 2000), although the latter do not see the MACv2-SP $N$ scalings, ensuring that anthropogenic aerosols do not exert a secondary indirect effect in the present study. HadGEM3 uses the Prognostic Cloud fraction and Prognostic Condensate scheme (PC2, Wilson et al., 2008) that simulates the mass-mixing ratios of water vapour, cloud liquid and ice, as well as the fractional cover of liquid, ice, and mixed-phase clouds.

The Norwegian Earth System Model (NorESM Bentsen et al., 2013; Iversen et al., 2013; Kirkevåg et al., 2013) uses the atmospheric component of the Oslo version of the Community Atmosphere Model (CAM4-Oslo), which differs from the original CAM4 (Neale et al., 2013) through the modified treatment of aerosols and their interaction with clouds (Kirkevåg et al., 2013). The model has a finite-volume dynamical core and the original version 4 of the Community Land Model (CLM4) of CCSM4 (Lawrence et al., 2011). NorESM uses the CAM-RT radiation scheme by Collins et al. (2006). Like ECHAM-HAM and ECHAM, NorESM sets all background aerosol emission to pre-industrial levels representative of 1850. These background conditions include sulphate from tropospheric volcanoes and from DMS, as well as organic matter from land and ocean biogenic processes, mineral dust and sea salt. Sea salt emissions are parameterised as a function of wind speed

and temperature (Struthers et al., 2011), while other pre-industrial aerosol emissions are prescribed following Kirkevåg et al. (2013). These are, in the case of NorESM, sulphate, organic matter and BC aerosols originating from fossil fuel emissions and biomass burning (Lamarque et al., 2010).

**Appendix B: Model diversity in cloud properties and surface albedo**

The model diversity in RF and ERF is larger when cloudy skies are considered. We therefore assess the model diversity in cloud properties and compare the model climatologies calculated from the simulations for the mid-2000s against observational climatologies from satellite products, listed in Table A1. The observational products provide an orientation for realistic values, although satellite retrievals also have caveats (e.g., Grosvenor et al., 2018). Moreover, we document the here-used surface albedos for illustrating both the regional differences and the model diversity.

**B1    Macroscopic cloud properties**

We first assess the cloud shortwave radiative effect at the top of the atmosphere ($F_{\mathrm{cld}}$), thus the cloud effect on the planetary albedo. The multi-annual global mean $F_{\mathrm{cld}}$ for $2001-2010$ from CERES Ed. 4 is -45.8 Wm$^{-2}$, i.e., less negative than in most models (Table A2). This behaviour indicates a tendency of the models to have too reflective clouds consistent with other model evaluations (Nam et al., 2012; Crueger et al., 2018, Lohmann and Neubauer, submitted). The spatial patterns of modelled $F_{\mathrm{cld}}$

are generally similar, but regionally the differences can be more distinct (Figure A1).

To better characterise the model diversity in clouds, we compare the simulated total cloud cover ($f$) and liquid water path ($l_{\mathrm{cld}}$) to satellite climatologies from ISCCP and MAC-LWP, respectively (Table A1). Most models underestimate both $f$ and $l_{\mathrm{cld}}$ over the oceans compared to the satellite retrievals, but having too few clouds does not necessarily imply too small amount of liquid or vice versa (Table A2). The spatial patterns (Figure A1) show a tendency of the models for underestimating $f$ in

the stratocumulus decks in the Southeastern regions of the Pacific and Atlantic Ocean, where aerosol-cloud interactions are thought to be important. The models, however, disagree on the values for $f$ and $l_{\mathrm{cld}}$ in those regions. Moreover, the models show a large diversity in $l_{\mathrm{cld}}$ in the extra-tropical storm tracks. NorESM shows the largest maximum $l_{\mathrm{cld}}$ exceeding 200 gm$^{-2}$. Our findings for $l_{\mathrm{cld}}$ are consistent with a similar regional comparison between HadGEM and CAM (Malavelle et al., 2017), the latter of which has a similar atmospheric component as NorESM (see Appendix A).

**B2    Cloud microphysical properties**

The reported differences in macroscopic cloud properties among the models raise the question how different the cloud droplet number concentrations ($N$) are. We find that the models show large diversity in the pattern of $N$ for present-day conditions as shown in Figure A2. Note that we show the mean in-cloud droplet number concentration, which means that regions without clouds are not included when averaging $N$. It is noteworthy that in the models $N$ is calculated for stratiform cloud types, but

can additionally include detrained droplets from anvils of deep convection. The spatial pattern of $N$ in ECHAM is not shown due to the simplistic treatment in the model. ECHAM employs statically prescribed values for $N$, which are constant with

height below 800 hPa and exponentially decrease aloft. The near-surface values in ECHAM are $N$=80 cm$^{-3}$ over ocean and $N$=180 cm$^{-3}$ elsewhere (not shown), and are multiplied with $\eta_N$ from MACv2-SP like in the other models.

Compared to the satellite product, the models typically underestimate $N$, e.g., in the stratocumulus decks, where also $f$ is underestimated. It remains an open question how much of the quantitative differences between the models and the satellite product is due to differences in the methods for diagnosing $N$ in the satellite retrievals and the models, but it is unlikely that the methods solely explain the diversity in the patterns of $N$. It is interesting that, despite these quantitative differences in $N$, the spatial pattern of $F_{\mathrm{cld}}$ compares reasonably well to observations (Figure A1), which might be a consequence of compensating differences from tuning the radiation balance at the top of the atmosphere. For instance, the behaviour of NorESM points to too much shortwave reflectivity by too thick clouds that overcompensate the missing reflection due to underestimated cloud cover.

## B3  Surface albedo

An additional influence on the radiative forcing of anthropogenic aerosol is the surface reflectivity for shortwave radiation. We therefore document the surface albedo for shortwave radiation from the participating models and the satellite product used in the offline radiative transfer calculations of this study. In the global mean, the models and the satellite product are very similar, with a surface albedo of 14$-$16%. However, the spatial distributions in Figure A3 indicate differences. The typical difference between less reflective ocean surfaces compared to land regions is apparent. Moreover, the analysis reveals diversity in the regional surface albedos of the participating models, typically related to areas affected by snow cover. Since such diversity in the surface albedo was already previously reported for aerosol-climate models with implications for the aerosol radiative forcing (e.g., Stier et al., 2007), future efforts are still needed for constraining the surface albedo in climate models.

*Author contributions.*  SF designed the study, performed the experiments with ECHAM, analysed the data of all models, and led the writing of the manuscript. SK performed the offline radiation-transfer calculations and compiled the surface albedo product MODIS-SSM/I. PR performed the experiments with NorESM, KH for ECHAM-HAM, NB for HadGEM, and TvN and DOD for EC-Earth. All authors contributed to the discussion of the results and the writing of the manuscript.

*Competing interests.*  The authors confirm that they have no competing interests.

*Acknowledgements.*  We thank the editor Hinrich Grothe for handling our manuscript and the three anonymous reviewers for their comments that helped improving the discussion article. This work is largely funded by the FP7 project "BACCHUS" (No. 603445). SF further thanks the Max Planck Society for funding. PS was additionally supported by the European Research Council (ERC) project "constRaining the EffeCts of Aerosols on Precipitation" (RECAP) under the European Union's Horizon 2020 research and innovation programme with grant agreement No. 724602 as well as by the Alexander von Humboldt Foundation. JM acknowledges the Academy of Finland for funding (No. 287440). We acknowledge the usage of the DKRZ supercomputer for running the simulations with ECHAM6.3. ECHAM6.3-

HAM2.3 simulations were performed through a grant from the Swiss National Supercomputing Centre (CSCS) under project ID: 652. We also acknowledge the usage of satellite data from the following providers. CERES data were obtained from the NASA Langley Research Center ordering tool ($http://ceres.larc.nasa.gov/$), ISCCP data from the International Satellite Cloud Climatology Project web site ($https://isccp.giss.nasa.gov$) maintained by the ISCCP research group at the NASA Goddard Institute for Space Studies, MAC-LWP data (Elsaesser et al., 2016) acquired as part of the activities of NASA's Science Mission Directorate, and archived and distributed by the Goddard Earth Sciences (GES) Data and Information Services Center (DISC, $https://disc.gsfc.nasa.gov$), and the cloud droplet number concentration climatology provided by the Vanderbilt University Institutional Repository ($https://ir.vanderbilt.edu/handle/1803/8374$). We thank Akos Horvath for providing information on MAC-LWP.

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

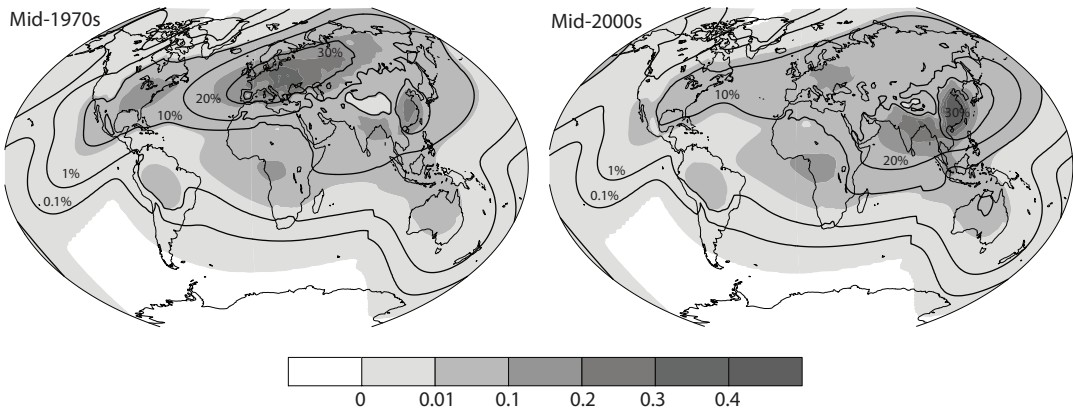

**Figure 1.** Mean anthropogenic aerosol optical depth ($\tau_a$, shaded) and fractional increase in cloud droplet number ($\eta_N$, contours) associated with anthropogenic aerosol. Shown are annual means of $\tau_a$ at 550 nm and $\eta_N$ for the (left) mid-1970s and (right) mid-2000s from MACv2-SP that prescribes annually repeating monthly maps of $\tau_a$ in the participating models. Note the non-linear scale.

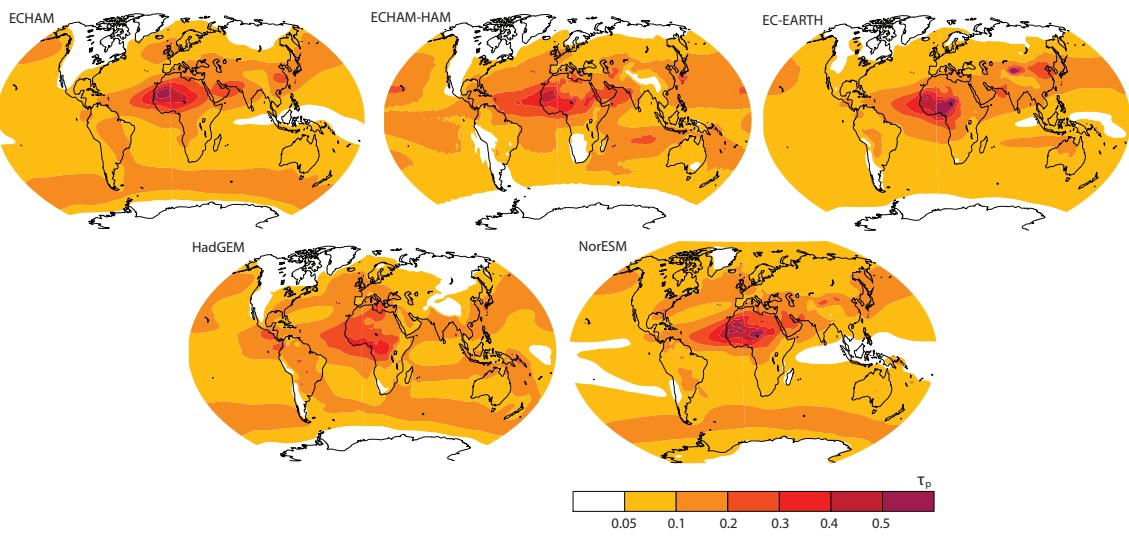

**Figure 2.** Mean pre-industrial aerosol optical depth ($\tau_p$). Shown are annual means of $\tau_p$ of the radiation band around 550 nm for each model.

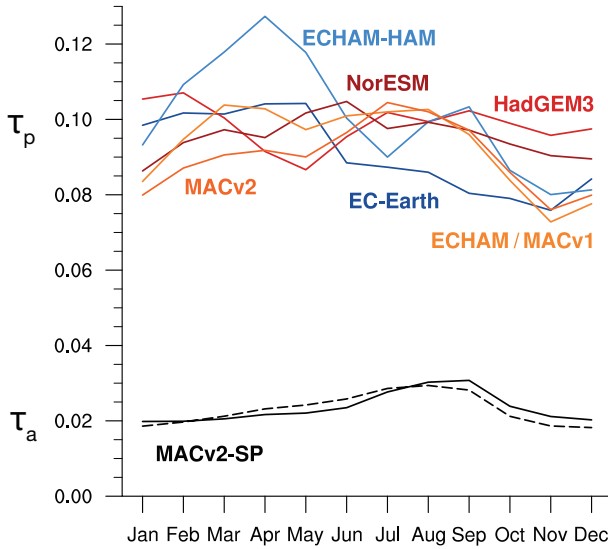

**Figure 3.** Annual cycle of the global mean aerosol optical depth at 550 nm. Shown are monthly means of (colors) $\tau_p$ from the models and (black) $\tau_a$ for the (dashed) mid-1970s and (solid) mid-2000s from MACv2-SP.

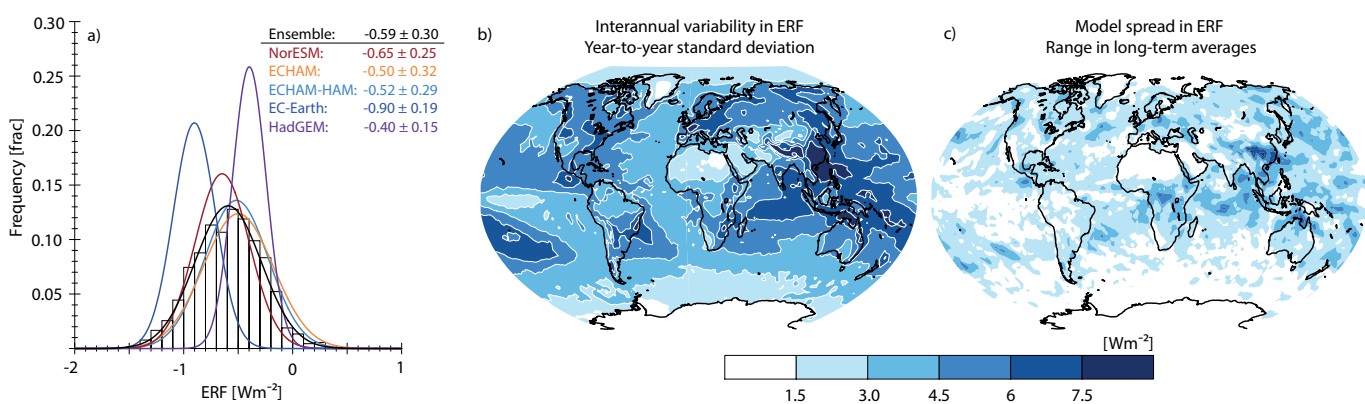

**Figure 4.** Variability in annual ERF estimates for the mid-2000s. Panel (a) shows Gaussian distributions of annual ERF estimates for present-day from (colors) individual model ensembles and (black) the entire multi-model, multi-member ensemble. The bars are the frequency histogram of one-year ERF estimates from all models, and the legend indicates the means and standard deviations of the ERF estimates. Panel (b) shows the regional standard deviation of annual contributions to ERF from the entire multi-model, multi-member ensemble as measure for the interannual variability internal to the model ensemble. Panel (c) shows the range in the long-term averaged ERFs of the models as measure for the spread in ERF associated with model differences. ERF are for the shortwave (SW) spectrum at the top of atmosphere (TOA) for all-sky conditions.

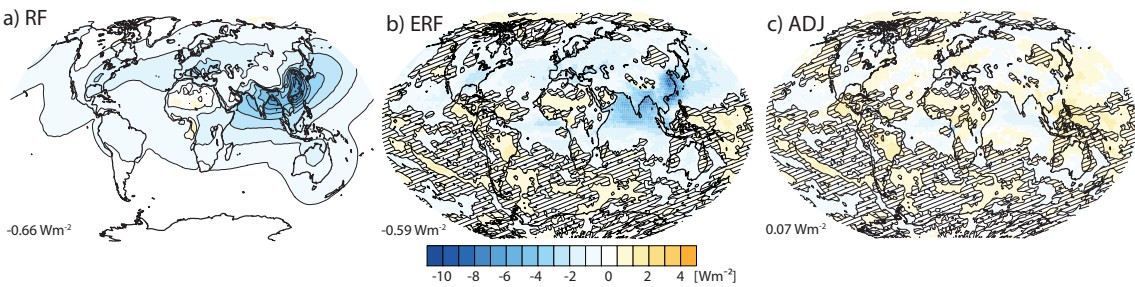

**Figure 5.** Multi-model, multi-member ensemble mean of the anthropogenic aerosol radiative effects for the mid-2000s. Shown are the (a) instantaneous and (b) effective radiative forcing as well as (c) the net contribution from rapid adjustments for SW at the TOA in all-sky conditions. Hatching in (b, c) indicates non-significant values at a 10% significance level. The numbers in the lower left corner are the spatial averages. The ensemble-mean RF is averaged over three climate models, the ensemble-mean ERF over five climate models, and the ensemble-mean adjustment is their difference.

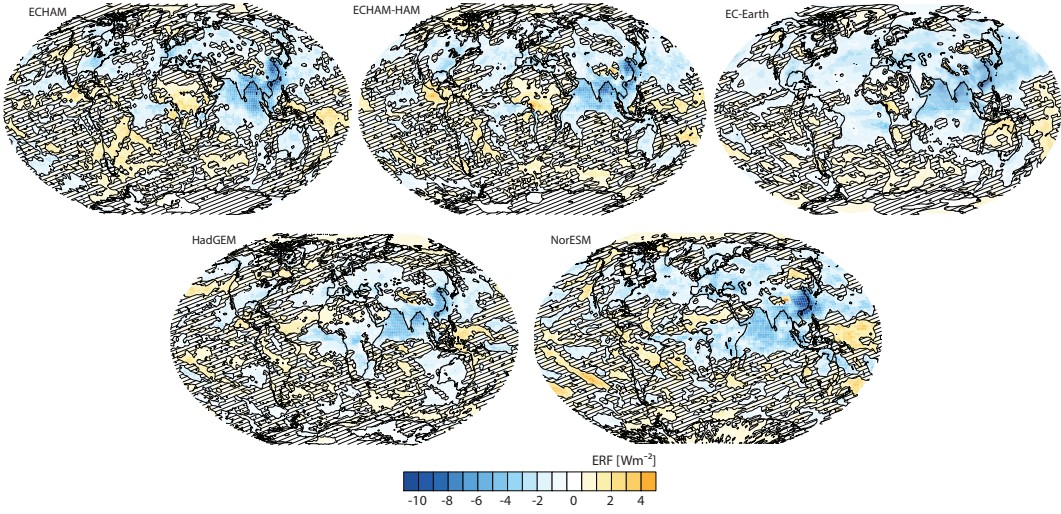

**Figure 6.** Multi-member ensemble mean of effective radiative effects of anthropogenic aerosol for the mid-2000s. Shown are the effective radiative forcing for SW at the TOA in all-sky conditions for each model. Hatching indicates non-significant values at a 10% significance level.

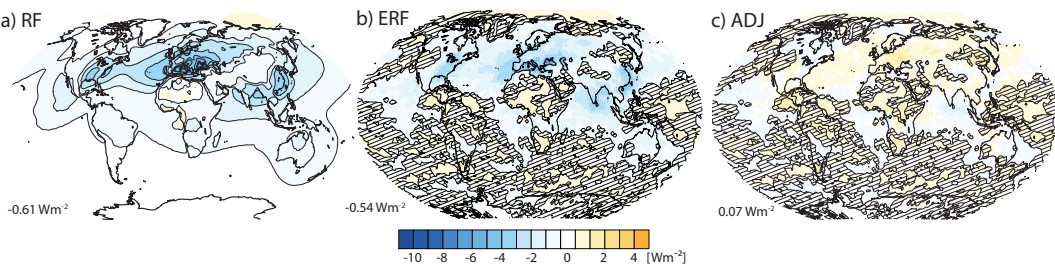

**Figure 7.** Multi-model, multi-member ensemble mean of the anthropogenic aerosol radiative effects for the mid-1970s. As Figure 5, but with the anthropogenic aerosol pattern of the mid-1970s.

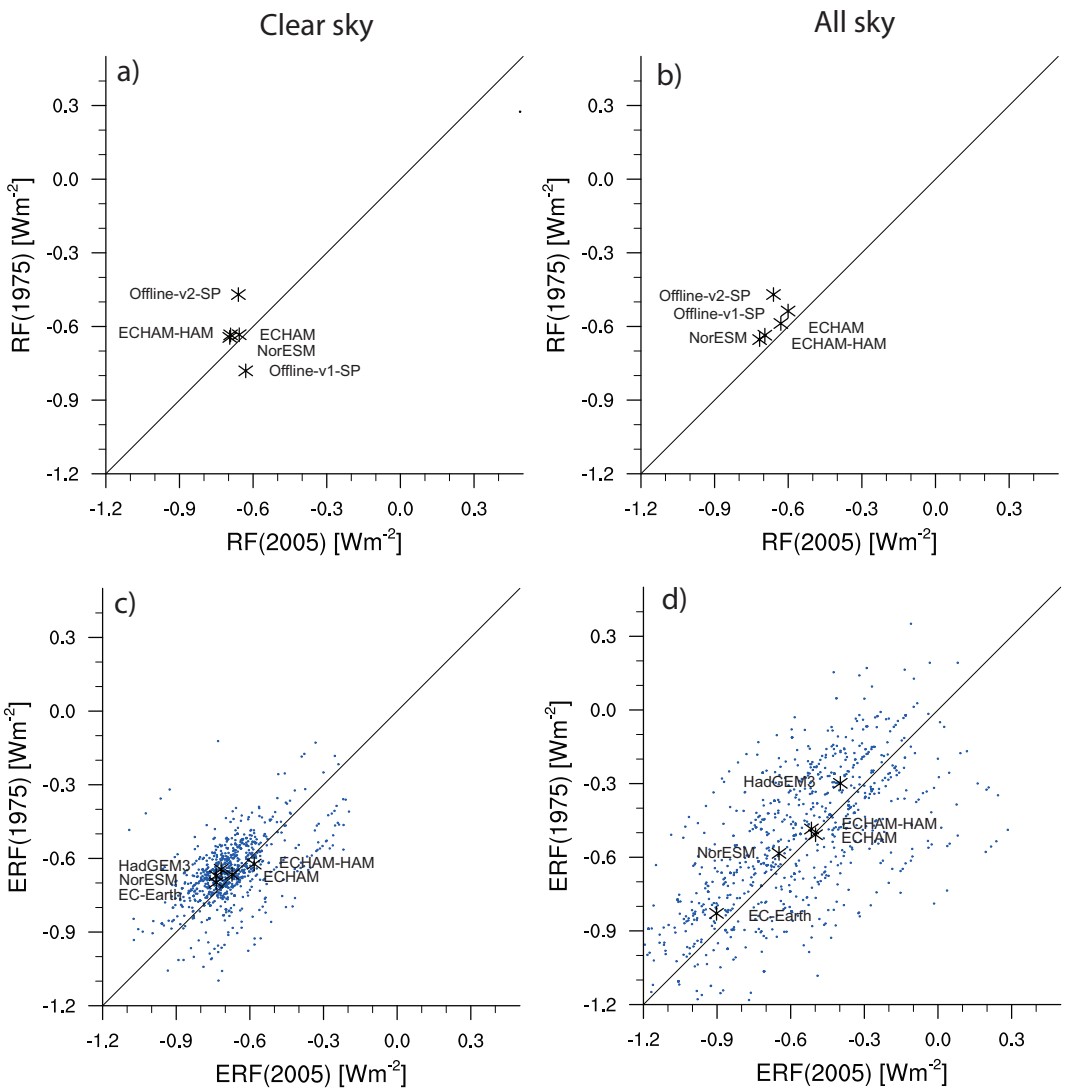

**Figure 8.** Anthropogenic aerosol forcing of the mid-1970s against the mid-2000s. Shown are the (top) instantaneous and (bottom) effective radiative forcing for SW at the TOA from the pollution of the mid-1970s against the mid-2000s for (left) clear and (right) all sky. Thick crosses are the ensemble means. Blue dots in (c, d) are the model averages of individual years representing the year-to-year variability internal to the model ensemble.

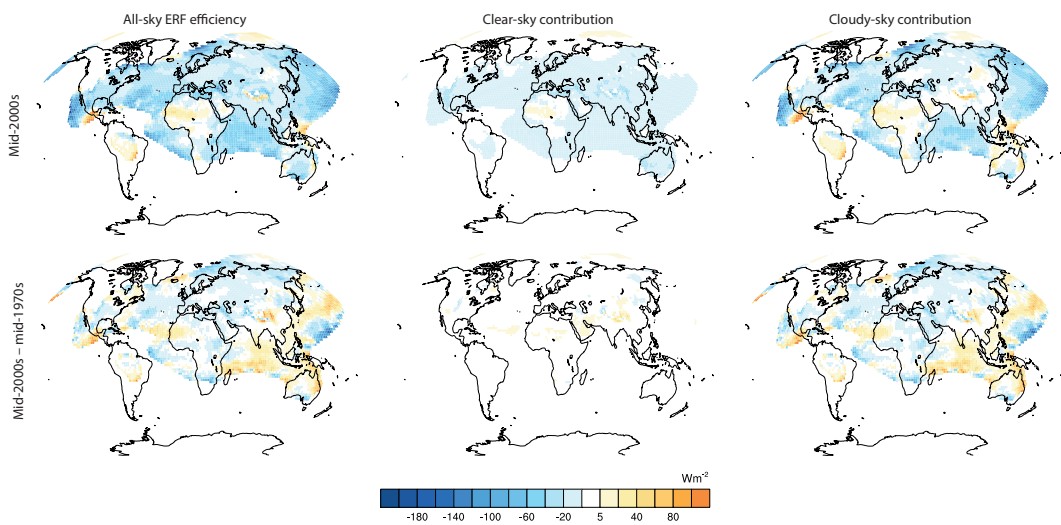

**Figure 9.** Anthropogenic aerosol effective radiative forcing efficiencies, in W m$^{-2}$ per unit optical depth, for (left) all-sky, (middle) clear-sky, and (right) cloudy-sky. The top row shows efficiencies for mid-2000s anthropogenic aerosols. The bottom row shows differences made by using the pattern for the mid-1970s.

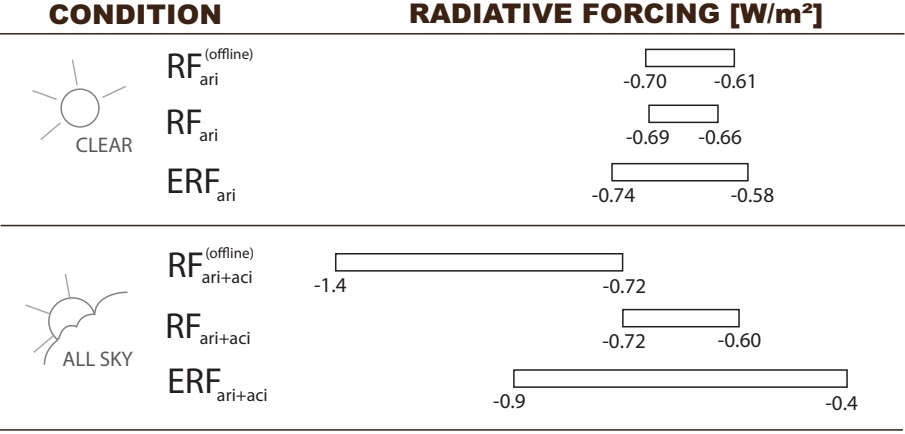

**Figure 10.** Summary of model spread in anthropogenic aerosol forcing for the mid-2000s. Shown are the instantaneous (RF) and effective radiative forcing (ERF) of aerosol-radiation and aerosol-cloud interactions for the shortwave spectrum at the top of the atmosphere for clear and all sky from Tab.2. The RF from the offline radiation-transfer calculations consider additional uncertainty sources and are shown as separate bars. Refer to Section 2.1 for details.

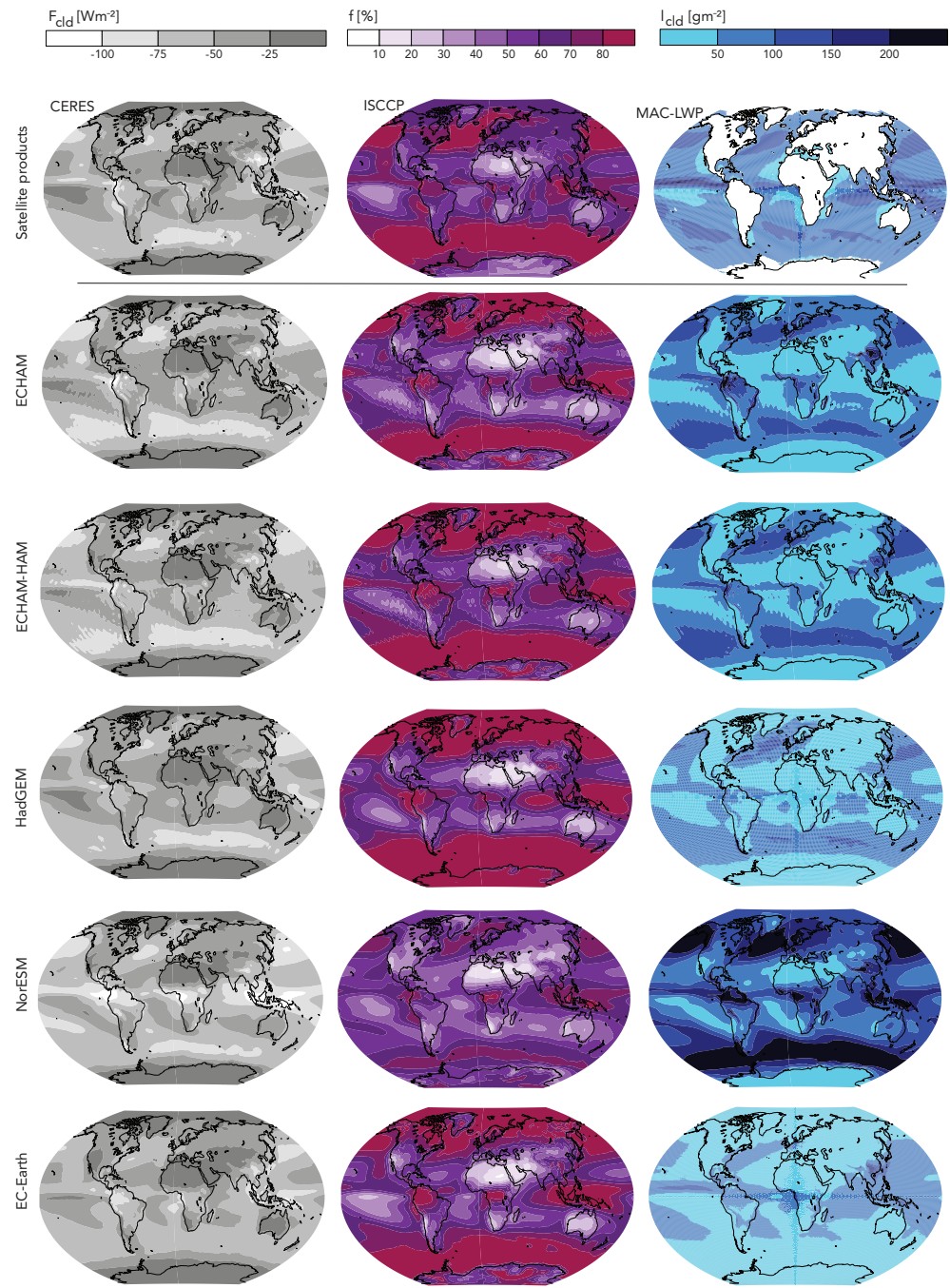

**Figure A1.** Multi-member ensemble means of cloud characteristics for the mid-2000s compared to climatologies derived from satellite observations (Table A1). Shown are the mean (left column) SW cloud radiative effect at the TOA, $F_{\mathrm{cld}}$, (middle column) total cloud cover, $f$, and (right column) liquid water path, $l_{\mathrm{cld}}$ from (top row) the satellite products and (rows beneath) the models. Areas without available data are shaded white.

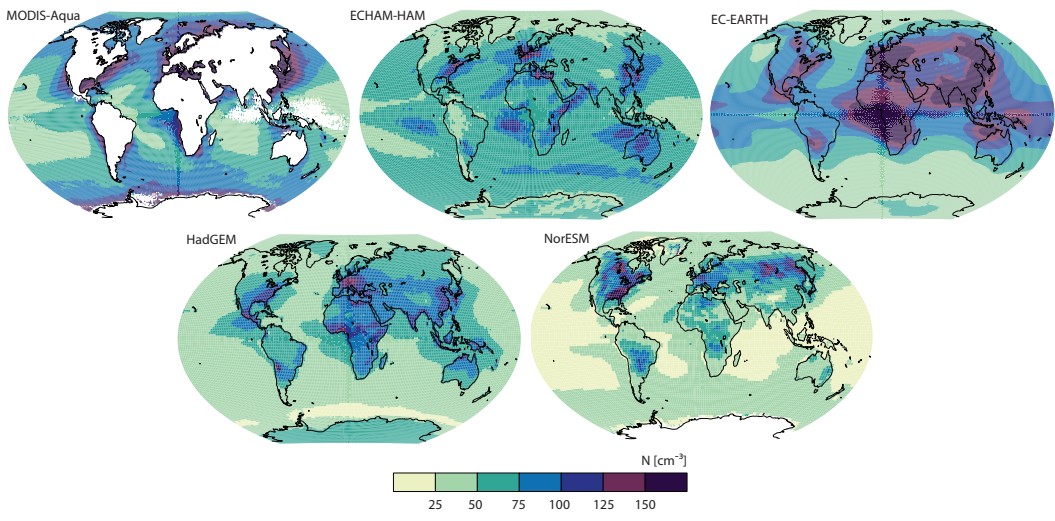

**Figure A2.** In-cloud droplet number concentration for the mid-2000s. Shown are the annually and vertically averaged in-cloud droplet number concentration ($N$) from the aerosol-climate models and from the MODIS satellite product by Bennartz and Rausch (2017). Areas without available data are shaded white.

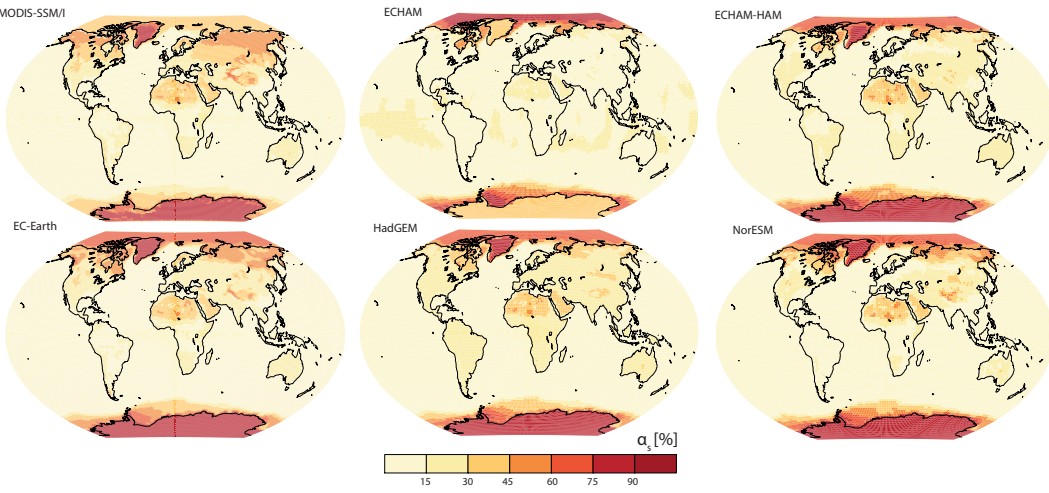

**Figure A3.** Surface albedo for shortwave radiation for the mid-2000s. Shown are the mean surface albedo for shortwave radiation ($\alpha_s$) from the models and the satellite product from Kinne et al. (2013).

**Table 1.** Model experimental setup

| Model | Horizontal resolution (longitude x latitude) | Number of vertical levels | Pre-industrial aerosol (1850) | Anthropogenic aerosol (increase since 1850) |
|---|---|---|---|---|
| ECHAM | 1.875° x 1.875° | 47 | MACv1 climatology | MACv2-SP |
| ECHAM-HAM | 1.875° x 1.875° | 47 | Online | MACv2-SP |
| EC-Earth | 1.875° x 1.875° | 91 | TM5 climatology | MACv2-SP |
| HadGEM3 | 1.875° x 1.25° | 85 | HadGEM3 climatology | MACv2-SP |
| NorESM | 2.5° x 1.894° | 26 | Online | MACv2-SP |
| Offline-v2-SP | 1° x 1° | 20 | MACv2 | MACv2-SP |
| Offline-v1-SP | 1° x 1° | 20 | MACv1 | MACv2-SP |
| Offline-v2 | 1° x 1° | 20 | MACv2 | MACv2 |
| Offline-v1 | 1° x 1° | 20 | MACv1 | MACv1 |

**Table 2.** Ensemble averages of the shortwave instantaneous (RF) and effective (ERF) radiative forcing, and net contribution from rapid adjustments (ADJ) at the surface (SFC) and the top of the atmosphere (TOA) for all sky (clear sky) in $\text{Wm}^{-2}$ for the period 1850 to 2005. The first block shows aerosol-climate models with MACv2-SP, and the second block shows estimates of the offline radiative transfer model.

| | $\text{RF}_{\text{SFC}}$ | $\text{RF}_{\text{TOA}}$ | $\text{ERF}_{\text{TOA}}$ | $\text{ADJ}_{\text{TOA}}$ |
|---|---|---|---|---|
| ECHAM | -1.52 (-1.64) | -0.60 (-0.66) | -0.50 (-0.67) | 0.1 (-0.01) |
| ECHAM-HAM | -1.63 (-1.67) | -0.72 (-0.69) | -0.52 (-0.58) | 0.2 (0.11) |
| EC-Earth | / | / | -0.90 (-0.74) | / |
| HadGEM3 | / | / | -0.40 (-0.72) | / |
| NorESM | -1.46 (-1.60) | -0.68 (-0.68) | -0.65 (-0.74) | 0.03 (-0.06) |
| Offline-v2-SP | -1.8 (-1.7) | -0.75 (-0.62) | / | / |
| Offline-v1-SP | -1.7 (-1.6) | -0.72 (-0.61) | / | / |
| Offline-v2 | -2.3 (-1.9) | -1.1 (-0.70) | / | / |
| Offline-v1 | -2.7 (-2.0) | -1.4 (-0.63) | / | / |

**Table A1.** Gridded climatologies of satellite retrievals used for model evaluation.

| Name | Description | Variable | Time |
|------|-------------|----------|------|
| CERES | Energy balanced and filled data of the Clouds and the Earth's Radiant Energy System, Ed. 4 (Loeb et al., 2009) | Cloud shortwave radiative effects at the top of the atmosphere, $F_{cld}$ [Wm$^{-2}$] | $2001-2014$ |
| ISCCP | International Satellite Cloud Climatology Project (Rossow and Schiffer, 1999) | Total cloud cover, $f$ [%] | $1983-2009$ |
| MAC-LWP | Multi-sensor Advanced Climatology (Elsaesser et al., 2016, 2017) | Liquid water path, $l_{cld}$ [gm$^{-2}$] | $2000-2016$ |
| MODIS | Climatology based on Moderate Resolution Imaging Spectroradiometer aboard Aqua (Bennartz and Rausch, 2017) | Cloud droplet number concentration in warm clouds, $N$ [cm$^{-3}$] | $2003-2015$ |
| MODIS-SSM/I | Climatology based on Moderate Resolution Imaging Spectroradiometer and microwave data (Kinne et al., 2013) | Surface albedo for shortwave radiation, $\alpha_s$ [%] | $1987-2007$ |

**Table A2.** Global mean statistics for clouds, aerosols and surface albedo. The numbers given for $l_{cld}$ and $N$ are averages over ocean regions, consistent with the satellite data availability (Figures A1 and A2). Details on the satellite products are listed in Table A1.

|  | $F_{cld}$ [Wm$^{-2}$] | $f$ [%] | $l_{cld}$ [gm$^{-2}$] | $N$ [cm$^{-3}$] | $\tau_p$ | $\alpha_s$ [%] |
|---|---|---|---|---|---|---|
| ECHAM | -47.5 | 63 | 65 | 84 | 0.093 | 16 |
| ECHAM-HAM | -49.1 | 68 | 69 | 65 | 0.097 | 15 |
| EC-Earth | -46.2 | 65 | 42 | 91 | 0.091 | 15 |
| HadGEM3 | -44.3 | 69 | 57 | 56 | 0.098 | 15 |
| NorESM | -55.5 | 55 | 133 | 34 | 0.096 | 14 |
| Satellite retrieval | -45.8 | 66 | 82 | 77 | - | 15 |