# Peer review of "Anthropogenic aerosol forcing - insights from multiple estimates from aerosol-climate models with reduced complexity"

_Atmospheric Chemistry and Physics, 2018_

## Referee Comment (RC1) · Anonymous Referee #2 · 26 Sep 2018

The manuscript presents a 4-model ensemble assessment of simulation variability for anthropogenic aerosol radiative forcing simulations. The four models represent a reasonable (if small) cross-section of the global models available. My main comments are focused on improving the clarity of analysis and presentation.

The estimate of variability in ERF seems to be overestimated: it is based on differentiating the time-series of pre-industrial simulations from those with anthropogenic aerosols. Should not an average of the pre-industrial simulations be used for the differencing baseline to avoid this? This is relevant to the discussion of inter-model variability relative to natural variability as well. Further, since the differences are done for each

of the three anthropogenically-influenced simulations, does it make sense to discuss correlations due to common variations driven by this approach?

I found it difficult to nail down exactly what was fixed between the different models in the simulations. Line 20 of page 2: ".. prescribing identical anth. aerosol optical properties across models allows us. . .. if we . . . know the aerosol distribution" - suggests that optical properties and concentrations are prescribed. Line 9 of page 3 indicates that they "prescribe identical optical properties of anthropogenic aerosols and an associate effect on the cloud reflectivity . . .. ", which I assume to mean only the intrinsic optical properties. However on page 5 , line 24, it appears, again, that the optical depth is prescribed (".. with pre-industrial aerosol optical depth. . . as of the year 1850, three experiments with with tau-p and anthropogenic aerosol from MACv2-SP for the year. . ..") , an extensive prescription that appears to fix also the emissions/atmospheric loads of the aerosol. This is fundamental to the paper and should be made crystal clear to the reader, especially in light of the findings about intra-model variability. For example, at line 19 of page 2, the point is made that "uncertainties in process modeling of anthropogenic aerosol" can be separated, but if optical depth is prescribed, I don't see how this is correct.

On numerous occasions, I was confused by wording and lack of specificity. I recommend that the authors perform a through line-by-line reading to make everything as clear as possible. Here are a few examples: 0) The term "multi-estimates" in the title does not appear to be widely used. Perhaps "multiple model estimates" might be more intuitive and familiar to the reader. 1 ) Abstract, line 4: "In those models we reduce. . ." - this makes it sound like a reference to only the models in the CMIP6. Better: "Here we reduce. . ." 2) Abstract, line 11 : "we reduce model diversity in clouds and use. . ." here "model diversity in clouds" is too vague - what is it referring to? 3) final sentence: what does "more stringent test" mean?

In Sec. 2.1, it is stated that anthropogenic aerosols are included in the pre-industrial burden, but don't form the majority contributor of AOD in the NorESM. However, the

reader needs more information about this to evaluate not the difference between anthropogenic and natural aerosols, but between pre-industrial and more contemporary simulations. One way to do this would be, for example, by providing the absolute anthropogenic contribution to global AOD in the two cases, to show if the pre-industrial case the anthropogenic contributions are small enough not to invalidate the results from this model relative to the others in the difference.

Last sentence of page 9: please provide some quantitative estimate of possible differences in natural emissions between pre-industrial and current day (for example due to land use changes etc.)

Line 17 of page 10: Clarity: it is not clear how consideration of variability does not affect an actual change in ERF. Perhaps the authors mean that they perceive the change as small relative to additional changes reflecting variability? This point is made more clearly in the conclusion.
* * *

---

## Referee Comment (RC2) · Anonymous Referee #1 · 5 Oct 2018

This manuscript examines the radiative forcing of anthropogenic aerosols in simulations with a small set of global models following the protocol for the Radiative Forcing MIP now in progress as part of CMIP6. The RFMIP aerosol specification, on which the lead authors were also a co-authors, provides a description of the anthropogenic aerosol in purely radiative terms i.e. as those parameters that enter the radiative transfer equation, and as their differential impact to cloud droplet number. Having eliminated model differences in what the aerosols are, the authors examine here how other model differences impact the radiative forcing. This could be considered a prototype for studies that might be done with the larger collection of RFMIP results when these become available. The authors report on the inter-model spread in effective radiative forcing

(ERF) at present-day, show differences in the present-day distribution of background clouds and aerosols, and examine how the shift in the aerosol distribution between the 1970s and present day has impacted the RF from anthropogenic aerosols.

This work is potentially interesting but not yet mature enough to publish. The work lacks an explicit motivating question, in the absence of which the variety of results presented is hard to interpret coherently. Some results, especially the off-line radiation calculations and the cursory comparison of model clouds and droplet number to observations, seem especially unconnected to the rest of the material. There are important methodological errors in how ERF is computed and in how the set of simulations is conceived of. Important opportunities for deeper understanding are also missed, especially in making connections between the background state of each model and the resulting diversity of ERF from anthropogenic aerosols. It is understandable that the lead authors wish to exploit something from the experiments they have helped design. The scientific community will nonetheless benefit more from work that exploits the simulations to answer specific questions.

Structure and focus:

What question do the authors seek to address in this work? One possibility would be "to what extent is the signal from anthropogenic aerosol detectable against the background of uncertainty and natural variability?" (I understand this to be one of the motivating questions of RFMIP although progress could be made without using formal detection and attribution machinery). Another would be "how does the background meteorological and/or aerosol state affect the radiative forcing of anthropogenic aerosols?"

In the absence of a clearly-articulated motivating question it is hard to know how to interpret results. One suspects that not all the material belongs in the same manuscript. If the goal is to understand the range of values of ERF that might be expected from the same aerosol across different models then the motivation for sections 3.3, 3.4, and 3.5 is unclear. If the question is understanding how background state affects ERF

then substantially more work will be required to link the quite cursory characterization of differences across models to the spread in ERF. Neither of these questions would motivate the also-cursory comparison of models and observations.

What is the intent of showing model-observation comparisons in section 3.3, or the offline radiation calculations in section 3.4? One might infer that the authors hope to address the ability to estimate real-world ERF from historical observations but this is not explained clearly.

Methodology:

Effective radiative forcing relates long-term radiative perturbations and long-term response. It does not make sense to look at yearly averages. The protocol for CMIP and RFMIP, following doi:10.1002/2016JD025320, is for 30-year simulations — precisely to average out model internal variability.

What motivates the use of multi-model means in 5-7, 9-10? An ensemble mean is the best estimate of the expectation value of some quantity when the samples are independent and uncorrelated, but this is unlikely to be the case in the small set of simulations here (or even in the larger collection to be collected through RFMIP).

Although the authors may well remove the comparisons to observations it is remiss to present inferences of drop number from satellites without mentioning the very many caveats around such estimates. See the careful review in doi:10.1029/2017RG000593.

Section 3.5 seems to illustrate that even a large spatial shift in aerosols has a relatively small impact on ERF. It's not clear why this bears mentioning - is there some surprise here? One might naively expect that the same aerosol burden would have roughly the same impact no matter where it was on the planet.

Smaller points:

The word "comparably" is used incorrectly in several places in the manuscript. The authors likely mean "relatively."

The introduction is so indirect as to be unclear. It would be better to start with motivating questions more specific to this study than "what is the anthropogenic aerosol forcing."

Far more detail is provided about each model than is useful. The only details that are really needed are those that might have bearing on interpreting the results presented here.

The simulations run from 2000-2010 but are treated as a statistically homogeneous set. Is this fair? It certainly deserves from comment.

In section 3,3 readers will appreciate a symbol for top-of-atmosphere shortwave cloud radiative effect that is not a capitalized version of the symbol for cloud fraction.

Do the conclusions in the last paragraph differ from the RFMIP protocol, or from community practice?
* * *

---

## Author Comment (AC1) · 27 Nov 2018

**Response to the referee comments for the manuscript:**
**"Anthropogenic aerosol forcing - insights from multi-estimates from aerosol-climate models with reduced complexity" by Fiedler et al.**

We thank the anonymous referees for their comments that helped improving the manuscript under discussion in Atmospheric Chemistry and Physics. Our main changes of the earlier version of the manuscript are:
(1) An improved presentation of our motivation with a revised introduction and introductory statements in the sections,
(2) More detailed explanations of our experiment, data and analysis strategy including improvements on statements on the reasons for computing the year-to-year variability in ERF as well as on model differences and similarities for improving the clarity of the text,
(3) new appendices for documenting model differences in the representation of physical processes and simulated cloud properties for improving the coherence and reading flow of the manuscript,
(4) And the extension of our model ensemble with the newly available EC-Earth experiments following our protocol.
Our replies are given in blue below the referee comments in black.

**Anonymous Referee #1**
This manuscript examines the radiative forcing of anthropogenic aerosols in simulations with a small set of global models following the protocol for the Radiative Forcing MIP now in progress as part of CMIP6. The RFMIP aerosol specification, on which the lead authors were also a co-authors, provides a description of the anthropogenic aerosol in purely radiative terms i.e. as those parameters that enter the radiative transfer equation, and as their differential impact to cloud droplet number. Having eliminated model differences in what the aerosols are, the authors examine here how other model differences impact the radiative forcing. This could be considered a prototype for studies that might be done with the larger collection of RFMIP results when these become available. The authors report on the inter-model spread in effective radiative forcing (ERF) at present-day, show differences in the present-day distribution of background clouds and aerosols, and examine how the shift in the aerosol distribution between the 1970s and present day has impacted the RF from anthropogenic aerosols. This work is potentially interesting but not yet mature enough to publish. The work lacks an explicit motivating question, in the absence of which the variety of results presented is hard to interpret coherently. Some results, especially the off-line radiation calculations and the cursory comparison of model clouds and droplet number to observations, seem especially unconnected to the rest of the material. There are important methodological errors in how ERF is computed and in how the set of simulations is conceived of. Important opportunities for deeper understanding are also missed, especially in making connections between the background state of each model and the resulting diversity of ERF from anthropogenic aerosols. It is understandable that the lead authors wish to exploit something from the experiments they have helped design. The scientific community will nonetheless benefit more from work that exploits the simulations to answer specific questions.

Thank you for your comments. Our work can be seen as a pilot study for RFMIP, where models use the MACv2-SP parameterisation of anthropogenic aerosol optical properties and associated change in the cloud droplet number concentration for assessing model errors in radiative transfer. It is important to underline that we only unify the treatment of anthropogenic aerosol, i.e., the natural aerosol is still model-dependent.

Our aim is an assessment of the impact of the spatial change of the anthropogenic aerosol between the mid-1970s and present-day as well as the role of model-internal variability with an ensemble of modern aerosol-climate models. We improve the presentation of our motivation and coherence of the analyses in the revised manuscript. For instance, we now state our research questions already in the second paragraph rather than at the end of the introduction. Please refer to our responses below for more details on the revision.

Structure and focus:

1. What question do the authors seek to address in this work? One possibility would be "to what extent is the signal from anthropogenic aerosol detectable against the background of uncertainty and natural variability?" (I understand this to be one of the motivating questions of RFMIP although progress could be made without using formal detection and attribution machinery). Another would be "how does the background meteorological and/or aerosol state affect the radiative forcing of anthropogenic aerosols?" In the absence of a clearly-articulated motivating question it is hard to know how to interpret results. One suspects that not all the material belongs in the same manuscript. If the goal is to understand the range of values of ERF that might be expected from the same aerosol across different models then the motivation for sections 3.3, 3.4, and 3.5 is unclear. If the question is understanding how background state affects ERF then substantially more work will be required to link the quite cursory characterization of differences across models to the spread in ERF. Neither of these questions would motivate the also-cursory comparison of models and observations.

We have moved our motivation and research question to the beginning of the article. The revised introduction names the motivation and research questions in the first two paragraphs:

"Despite decades of research on the radiative forcing of anthropogenic aerosol, quantifying the present-day magnitude and reconstructing the historical evolution of the forcing remains challenging. Recent work has indicated that natural variability affects estimates of the effective radiative forcing (ERF) of anthropogenic aerosol (Fiedler et al., 2017). More specifically, natural variability was identified as a cause for increases and decreases in the global mean ERF associated with the spatial change in anthropogenic AOD ($\tau_a$) between the mid-1970s and mid-2000s. The anthropogenic aerosol pollution in the mid-1970s was herein larger in Europe and North America than in East Asia, whereas the opposite is the case in the mid-2000s. In addition to these regional changes in aerosol pollution, differences in the surface albedo, insolation, and cloud regimes between the aerosol transport regions of the Pacific and continental Europe may result in changes in the global ERF over time.

In light of model uncertainties (e.g., Kinne et al.,2006, Quaas et al., 2009, Lohmann et al., 2010, Lacagnina et al., 2015, Koffi et al., 2016), a single model as used in Fiedler et al. (2017) does not necessarily represent the full spectrum of possible anthropogenic aerosol forcings. In the present study, we therefore revisit the question of Fiedler et al. (2017): "Does the substantial spatial change of the anthropogenic aerosol between the mid-1970s and mid-2000s, reflected by the change in $\tau_a$ shown in Fig. 1, affect the global magnitude of ERF?" using ensembles of simulations from five global aerosol-climate models with reduced aerosol complexity. In this context, we additionally ask: "What is the relative contribution of variability amongst and within models to the spread in ERF?", and document the model diversity for the pre-industrial aerosol and cloud characteristics that are relevant for ERF of anthropogenic aerosol. Such model differences have previously been identified for other climate models (e.g., Nam et al., 2012, Fiedler et al., 2016, Crüger et al., 2018)."

2. What is the intent of showing model-observation comparisons in section 3.3, or the offline radiation calculations in section 3.4? One might infer that the authors hope to address the ability to estimate real-world ERF from historical observations but this is not explained clearly.

We show the observations as an orientation for realistic values for model validation. Please note that Section 3.3 has been moved to the appendix for improving the reading flow of the article.

We state in the revised introduction: "We provide observational benchmarks for the inter-comparison of the complex models with satellite data and results from a stand-alone atmospheric radiation transfer model for quantifying differences in the instantaneous

radiative forcing (RF)",  in Appendix B (former Section 3.3):  "The model diversity in RF and ERF is larger when cloudy skies are considered. We therefore assess the model diversity in cloud properties and compare the models against observational climatologies from satellite products, (…). The observational products herein provide an orientation for realistic values, (…).", and at the beginning of Section 3.3 (former Section 3.4): "We use offline radiation transfer calculations for providing benchmarks for the instantaneous radiative forcing (RF) of the complex models. "

Methodology:
3. Effective radiative forcing relates long-term radiative perturbations and long-term response. It does not make sense to look at yearly averages. The protocol for CMIP and RFMIP, following doi: 10.1002/2016JD025320, is for 30-year simulations precisely to average out model internal variability.

We agree, it is precisely one of our points and important for later ERF analyses from CMIP6 simulations, i.e., we need to average over sufficiently long time periods for estimating ERF of a model. Fiedler et al. (2017) discuss the precision of ERF estimates from one climate model that depends on the confidence level, the magnitude of model internal variability and the number of years for averaging. Here, we show that the year-to-year standard deviation in ERF is similar to the model in Fiedler et al. (2017), i.e., the precision estimates are applicable to the here-used models, too. Short model simulations covering a few years, like studies have done in the past, are not suitable for calculating ERF and can lead to misleading results. It is important to keep this in mind for diagnosing ERF in transient climate experiments. e.g., by following the RFMIP recommendation of using three member ensembles with ten-year averages for time-varying ERF estimates.

In addition to our explanation in the last paragraph of Section 3.1, we now add in Section 2.2: "This approach is chosen for illustrating the effect of year-to-year variability on ERF estimates. (…) the RFMIP protocol recommends a thirty-year average for diagnosing the ERF of a model (Pincus et al., 2016)" and in the conclusion: "For instance, the protocol of RFMIP requests thirty-year averages for estimating the present-day ERF and three-member ensembles with ten-year averages for diagnosing decadal changes in ERF (Pincus et al., 2016)."

4. What motivates the use of multi-model means in 5-7, 9-10? An ensemble mean is the best estimate of the expectation value of some quantity when the samples are independent and uncorrelated, but this is unlikely to be the case in the small set of simulations here (or even in the larger collection to be collected through RFMIP).

The multi-model mean is useful for comparing individual model results to the same reference. We add in Section 3.1: "For doing so, we first calculate the multi-model mean as a reference value."

5. Although the authors may well remove the comparisons to observations it is remiss to present inferences of drop number from satellites without mentioning the very many caveats around such estimates. See the careful review in doi:10.1029/2017RG000593.

We agree that satellite retrievals are uncertain themselves and add in the Appendix (former Section 3.3): "The observational products herein provide an orientation for realistic values, although satellite retrievals also have caveats (e.g., Grosvenor et al. 2018)." The section on the cloud inter-comparison has been moved to the Appendix for improving the reading flow of the article.

6. Section 3.5 seems to illustrate that even a large spatial shift in aerosols has a relatively small impact on ERF. It's not clear why this bears mentioning - is there some surprise here? One might naively expect that the same aerosol burden would have roughly the same impact no matter where it was on the planet.

It is not obvious that the same change in global mean aerosol optical depth gives the same global ERF. We revise the introduction to make this clearer (refer to our reply to the first point). Additionally, we state at the beginning of Section 3.4 (former Section 3.5): "We assess the effect of a substantial spatial change of the $\tau_a$ maxima from Europe and the U.S. to East Asia between the mid-1970s and mid-2000s. One can additionally argue that the spatial differences in cloud regimes, insolation and surface albedo contribute to regionally different radiative effects resulting in a changing global ERF."

Smaller points:
7. The word "comparably" is used incorrectly in several places in the manuscript. The authors likely mean "relatively."

Replaced.

8. The introduction is so indirect as to be unclear. It would be better to start with motivating questions more specific to this study than "what is the anthropogenic aerosol forcing."

We revised the introduction. Please refer to our reply to your first point.

9. Far more detail is provided about each model than is useful. The only details that are really needed are those that might have bearing on interpreting the results presented here.

We focus on model differences in the pre-industrial aerosol and clouds that are relevant to the results on radiative forcing. For the sake of brevity, we have moved the overview on the model physics packages to the appendix and refer to it in Section 2.2: "We therefore keep for instance the model diversity for the physical parameterisations of radiation and clouds (Appendix A)" and add in the same section: "All other aspects remain model-dependent, e.g., the treatment of the pre-industrial aerosol and clouds (Appendix A)" and describe the model differences for the pre-industrial aerosol optical depth in a new paragraph: "We do not prescribe the same natural aerosol nor interfere with any other model components than prescribing the optical properties of anthropogenic aerosols and $\eta_N$. For instance, the pre-industrial aerosol optical depth ($\tau_p$) depends on the model (Fig. 2 and 3). Regional differences occur primarily over oceans and deserts, where observations are typically sparse. It is herein noteworthy that ECHAM-HAM runs with interactive parameterisations for dust and sea-salt aerosol resulting in different spatio-temporal variability in $\tau_p$ (Fig. 3) compared to the monthly mean climatology MACv1 in ECHAM. In the interactive parameterisations, the natural aerosol emissions, transport and deposition rely on meteorological processes that are difficult to represent in coarse-resolution climate models, e.g., desert-dust emissions strongly depend on the model representation of near-surface winds (e.g., Fiedler et al., 2016) such that constraining the desert-dust burden remains challenging in bottom-up aerosol modelling (e.g., Räisänen et al., 2013, Evan et al., 2014, Huneeus et al., 2016). "

10. The simulations run from 2000-2010 but are treated as a statistically homogeneous set. Is this fair? It certainly deserves from comment.

We add in Section 2.2.: "The first year of each 11-year run is considered as a spin-up period and is excluded from the analysis, thus all analyses are for the period 2001-2010. We have chosen the ten-year period for including variability in the boundary conditions."

11. In section 3,3 readers will appreciate a symbol for top-of-atmosphere shortwave cloud radiative effect that is not a capitalized version of the symbol for cloud fraction.

We remove the subscript in the symbol for the cloud fraction in the revised manuscript.

12. Do the conclusions in the last paragraph differ from the RFMIP protocol, or from community practice?

Past community practices partly differed from what is recommended in the RFMIP protocol and tested in the framework of our article. We have added: "The protocol of RFMIP requests thirty year averages for estimating the present-day ERF and three-member

ensembles with ten-year averages for diagnosing decadal changes in ERF (Pincus et al., 2017)."

**Anonymous Referee #2**
The manuscript presents a 4-model ensemble assessment of simulation variability for anthropogenic aerosol radiative forcing simulations. The four models represent a reasonable (if small) cross-section of the global models available. My main comments are focused on improving the clarity of analysis and presentation.

Thank you for your comments. We now additionally include EC-Earth experiments for a larger ensemble of five complex aerosol-climate models. We have worked on the language and added details throughout the manuscript for improving the clarity. Please refer to our more detailed responses below.

13. The estimate of variability in ERF seems to be overestimated: it is based on differentiating the time-series of pre-industrial simulations from those with anthropogenic aerosols. Should not an average of the pre-industrial simulations be used for the differencing baseline to avoid this? This is relevant to the discussion of inter-model variability relative to natural variability as well.

We define variability in ERF internal to the models as year-to-year variability, i.e., we compute annual means of the radiation budget for determining ERF. We herein subtract years with identical boundary conditions in the simulation without anthropogenic aerosol from the simulation with anthropogenic aerosol for each model. Using a mean of just the pre-industrial simulation would compute a yearly anomaly that would be different from what we define here as year-to-year variability.

In addition to our explanation in the last paragraph of Section 3.1, we now add in Section 2.2: "This approach is chosen for illustrating the effect of year-to-year variability on ERF estimates. (…) the RFMIP protocol recommends a thirty-year average for diagnosing the ERF of a model (Pincus et al., 2016)" and in the conclusion: "For instance, the protocol of RFMIP requests thirty-year averages for estimating the present-day ERF and three-member ensembles with ten-year averages for diagnosing decadal changes in ERF (Pincus et al., 2016)."

14. Further, since the differences are done for each of the three anthropogenically-influenced simulations, does it make sense to discuss correlations due to common variations driven by this approach? I found it difficult to nail down exactly what was fixed between the different models in the simulations. Line 20 of page 2: ".. prescribing identical anth. aerosol optical properties across models allows us: : :. if we : : : know the aerosol distribution" - suggests that optical properties and concentrations are prescribed. Line 9 of page 3 indicates that they "prescribe identical optical properties of anthropogenic aerosols and an associate effect on the cloud reflectivity : : :. ", which I assume to mean only the intrinsic optical properties. However on page 5 , line 24, it appears, again, that the optical depth is prescribed (".. with pre-industrial aerosol optical depth: : : as of the year 1850, three experiments with with tau-p and anthropogenic aerosol from MACv2-SP for the year: : :."), an extensive prescription that appears to fix also the emissions/atmospheric loads of the aerosol. This is fundamental to the paper and should be made crystal clear to the reader, especially in light of the findings about intra-model variability. For example, at line 19 of page 2, the point is made that "uncertainties in process modeling of anthropogenic aerosol" can be separated, but if optical depth is prescribed, I don't see how this is correct.

The revised introduction states: "Here, we prescribe observationally constrained optical properties of anthropogenic aerosol and an associated effect on the cloud droplet number concentration (…), but keep the full model diversity in other aspects. It allows us to eliminate the uncertainties in process modelling of anthropogenic aerosol and focus on the uncertainties in other processes influencing the radiative forcing. In other words, prescribing identical anthropogenic aerosol optical properties and an associated effect on the cloud droplet number concentration across models allows us to study those sources of uncertainty that remain if we pretend to know the spatial distribution of anthropogenic aerosol. We can thereby quantify the sole impact of other model differences, such as the

natural aerosol, meteorology, radiative transfer, and surface albedo, on the radiative forcing of observationally constrained anthropogenic aerosol in a state-of-the-art multi-model context.", we further add in Section 2.1: "All other aspects remain model-dependent, e.g., the treatment of the pre-industrial aerosol and clouds (Appendix A)" and document the model differences for the pre-industrial aerosol optical depth in a new paragraph: "We do not prescribe the same natural aerosol nor interfere with any other model components than prescribing the optical properties of anthropogenic aerosols and $\eta_N$. For instance, the pre-industrial aerosol optical depth ($\tau_p$) depends on the model (Fig. 2 and 3). Regional differences occur primarily over oceans and deserts, where observations are typically sparse. It is herein noteworthy that ECHAM-HAM runs with interactive parameterisations for dust and sea-salt aerosol resulting in different spatio-temporal variability in $\tau_p$ (Fig. 3) compared to the monthly mean climatology MACv1 in ECHAM. In the interactive parameterisations, the natural aerosol emissions, transport and deposition rely on meteorological processes that are difficult to represent in coarse-resolution climate models, e.g., desert-dust emissions strongly depend on the model representation of near-surface winds (e.g., Fiedler et al., 2016) such that constraining the desert-dust burden remains challenging in bottom-up aerosol modelling (e.g., Raisanen et al., 2013, Evan et al., 2014, Huneeus et al., 2016). ", and in Section 2.2: "Moreover, each participating model was free to individually set up all other aspects than the anthropogenic aerosol treatment. We therefore keep for instance the model diversity for the physical parameterisations of radiation and clouds (Appendix A)." The model diversity for clouds is documented in the appendix in the revised manuscript.

15. On numerous occasions, I was confused by wording and lack of specificity. I recommend that the authors perform a through line-by-line reading to make everything as clear as possible.
We have worked on the text and made the following changes in response to your examples:

16. Here are a few examples:
0) The term "multi-estimates" in the title does not appear to be widely used. Perhaps "multiple model estimates" might be more intuitive and familiar to the reader.
Changed to: "multiple estimates"

1 ) Abstract, line 4: "In those models we reduce: : :" - this makes it sound like a reference to only the models in the CMIP6. Better: "Here we reduce: : :"
Changed to: "We calculate the instantaneous radiative forcing (RF), effective radiative forcing (ERF), and rapid adjustments by comparing 10-year long ensemble simulations with aerosol distributions for 1850, the mid-1970s and the mid-2000s. The complexity of the anthropogenic aerosol is herein reduced"

2) Abstract, line 11 : "model diversity in clouds and use: : :" here "model diversity in clouds" is too vague - what is it referring to?
We removed the statement in the abstract and document the model differences in cloud droplet number, cloud cover, cloud radiative effects and cloud liquid water in the new appendix that we created in response to reviewer #1

3) final sentence: what does "more stringent test" mean?
Changed to: "better test"

17. In Sec. 2.1, it is stated that anthropogenic aerosols are included in the pre-industrial burden, but don't form the majority contributor of AOD in the NorESM. However, the reader needs more information about this to evaluate not the difference between anthropogenic and natural aerosols, but between pre-industrial and more contemporary simulations. One way to do this would be, for example, by providing the absolute anthropogenic contribution to global AOD in the two cases, to

show if the pre-industrial case the anthropogenic contributions are small enough not to invalidate the results from this model relative to the others in the difference.

> We have calculated the contributions of the anthropogenic AOD in 1850 in NorESM and add in the description of NorESM: "The 1850's global-mean $\tau_p$ in NorESM is 0.096, to which anthropogenic fossil-fuel emissions make a contribution of 0.002. For comparison, the year 2005 global-mean $\tau_a$ for MACv2-SP aerosols is 0.029.". This Section has moved to a new Appendix A in response to reviewer #1.

18. Last sentence of page 9: please provide some quantitative estimate of possible differences in natural emissions between pre-industrial and current day (for example due to land use changes etc.)

> We add: "Quantitative changes in natural aerosol burden between the pre-industrial and present-day remain unconstrained, e.g., model estimates of the anthropogenic fraction of desert dust are 10-60% associated with changes in land use and climate (Mahowald and Luo, 2003; Tegen et al., 2004; Stanelle et al., 2014)."

19. Line 17 of page 10: Clarity: it is not clear how consideration of variability does not affect an actual change in ERF. Perhaps the authors mean that they perceive the change as small relative to additional changes reflecting variability? This point is made more clearly in the conclusion.

> Replaced with: "The ensemble-averaged change in ERF is small relative to natural year-to-year variability in modelled ERFs (…)."

---

## Referee Report (RR1)

I have reviewed the revised version of Fiedler et al. Since I was not part of the first round of review, I have mostly restricted myself to determining whether the authors have addressed the comments raised by the other reviewers. However, I am a bit disappointed that those reviewers did not bring up the most poignant criticism of the "simple plumes" (SP) parameterization of aerosol–cloud interactions (ACI), which is that the sporadic transport of anthropogenic pollution into usually very clean regions is underrepresented.

My biggest concern with the manuscript is one raised by both of the original reviewers, which is that the manuscript is difficult to read. This is in part because vague phrasing abounds. One of the reviewers explicitly asked the authors to go through the text line by line to rephrase and reduce potential for confusion. That the authors have made only cursory improvements in the revision is disrespectful of the reviewers (and the manuscript's future readers). It also made me feel antagonistic enough that I seriously considered recommending rejection before deciding on major revisions; after all, what assurance do I have that the authors will consider my recommendations any more than the original reviewers'?

**1   Major organizational issues**

That being said, I also disagree with multiple comments made by the original reviewer who recommended rejection. While the manuscript currently reads like a grab bag of resultlets, none of which is fully developed into an interesting conclusion (as the original reviews pointed out), a good-faith effort at reorganizing the content, as well as rewriting to remove platitudes and vagueness, would lead to a manuscript well worth reading. I have collected a few suggestions for results to emphasize and discuss in greater depth:

- On the subject of the averaging required for reliable forcing estimates, have the authors considered the "nudging" approach, where the large-scale dynamics of the model is constrained to reanalysis (e.g., Zhang et al., 2014, ACP)? Can nudging shorten the integration periods or reduce the averaging required to make reliable estimates of transient forcing time evolution? What are the drawbacks or trade-offs that need to be considered?

- The point that it may be more fruitful to calculate forcing differences between different levels of anthropogenic pollution rather than a forcing relative to a poorly characterized "preindustrial" state is well taken. I should note that I do not understand why this point appears in the section it appears in, "Benchmarking RF" (but I also do not understand what "benchmarking" means as used by the authors).

- I believe the original reviewer's comment about the spatial shift in pollution from Europe/North America to Asia is wrong;[1] contrary to that reviewer's opinion, the fact that the AOD is similar in both time periods does not imply that the ERF or RF should necessarily be similar.
* * *
[1]Parenthetically, I also think the reviewer comment about ensemble means is incorrect; the sample mean is the optimal estimate of the population mean regardless of the caveats the reviewer lists. There is absolutely no guarantee that it will converge on the true forcing, but that is an inherent problem in modeling and unrelated to the number of models that participate.

However, for this finding to be illuminating, the authors should endeavor to explain why this is the case rather than simply stating the fact in Sec. 3.4. (As a side note, the numbers on p. 8 l. 26 are slightly different from the original manuscript; why is that?) The same holds for the discussion of the differences in efficacy. Why is the efficacy different between the time periods? Why do the efficacies increase in the less polluted regions? I would assume this is because the ACI sensitivity saturates. If so, what does that imply for the reliability of the SP method, where the model sees essentially the same average concentration of anthropogenic pollution, while in model configuration where the model is allowed to do its own transport, anthropogenic aerosol can sporadically intrude into clean, and therefore highly sensitive, regions? (I would say it indicates that the SP method will lead to a significant underestimate of the RFaci, but my point here is that the authors need to discuss their findings.)

- Continuing on the previous point, in the introduction, the authors say that one of their research questions is how differences in surface albedo, insolation, and cloud regimes affect ERF over time. However, they do not return to this question in the manuscript. If they follow my suggestion in my previous point, it will have the cobenefit of making their introduction more reflective of the paper.

**2 Clarity of writing**

Regarding clarity of the writing, one of the imprecisions that was a constant irritant was the definition of $F_{aci}$: I think it might mean ERFaci for EC-Earth and RFaci for the other models, but I still haven't been able to figure it out for sure. As for $F_{ari}$, I'm pretty sure that is the ERFari, from the description on page 4. But if $F_x$ refers to ERF for $x$ = ari and RF for $x$ = aci, that is extremely confusing.

The original reviewers went to some trouble to identify other particularly unclear passages. I am somewhat deterred by the lack of response by the authors, so I will not expend effort on listing further instances. Just by way of example, in the first paragraph of the conclusions:

- What does "the" in "the five state of the art models" mean? It makes it sound like this is an exhaustive list, so no other models are state of the art. I know that this is not what the authors mean, but the sloppy writing is doing them a disservice.

- "reflecting both natural variability and model differences affecting ERF" is such an unclear way of restating the previous clause in that sentence that it took me forever to figure out that is was meant as a restatement. Writing something like "reflecting that natural variability and model differences both contribute to the model diversity in ERF" would have made it clear immediately. I know that it is hard to identify unclear passages in something that one has written oneself, but this paper has 12 authors, so there was no shortage of opportunity for someone to approach the text in the role of an uninitiated reader.

- What are the "best" model-mean estimates? (For that matter, what does "model-mean" mean in that sentence?)

I do implore the authors to follow the suggestion of having someone (who will receive better compensation than a reviewer) go through the manuscript line by line to improve the writing.

**3   Minor comments**

- p. 5, l. 32 onward: how can you tell this is not just coincidence?

- p. 6, l. 28: I do not understand this sentence; what does "more than one model ensemble" refer to? More than an ensemble of runs from one model? More than the multi-model ensemble in this study?

- p. 7, l. 1: "are" → "is"

- The word "herein" appears frequently, and I don't think I understood what it was supposed to mean once

- I like the appendices, and I do not understand why the original reviewer complained about "too much detail" in the model description (the "too many notes, Mr. Mozart" line from *Amadeus* comes to mind). I believe the long form of "DMS" is two words (p. 10, l. 29). While "vertically integrated liquid water content" (p. 12, l. 25) is correct, why not call it by its better known name, liquid water path?

- In the author contributions, "lead" → "led"

- In Tab. 3, what does it mean when a clear-sky ERF is more negative than a cloudy-sky ERF? Positive forcing by ACI?

---

## Referee Report (RR2)

I commend the authors on a much improved manuscript. The significant investment of effort into a sentence-by-sentence rewriting of the manuscript by the entire author list has made the conclusions much easier for the reader to grasp. As a minor comment, since "larger audience" is mentioned multiple times in the authors' reply, I do want to clarify that the issue was never that the reader had to be an expert, but rather that even the experts were left confused.

My very minor **comments use the differences document attached to the authors' responses** because this was the only document I had access to at the time due to computer problems while traveling.

p. 4, l. 10: I am still unclear on the definition of $F_{ari}$ (and aci). From the author replies, I am led to believe these are not radiative flux differences, but rather ... what? The physical processes themselves?

p. 5, l. 25: what does "interpretation" mean?

p. 6, l. 18: I know what you mean, but you might want to rewrite this sentence to make it easier to understand why this is efficient; on the previous line, "three experiment" → "three experiments"

p. 7, l. 13: state explicitly that "ADJ" is the semidirect effect except for EC-EARTH. I am a little confused that the adjustments are positive; I thought that they were typically negative in GCMs. Is this because the Twomey effect is weaker than the direct aerosol effect in MAC–SP (which I think is also unusual)? It might be good to provide a few sentences of context for these MAC–SP results compared to other multi-model ensembles (AeroCom, CMIP5); I think most readers will be convinced that they should expect lower intermodel spread when the Twomey effect is prescribed, but they may share my surprise when effects change sign compared to the conventional wisdom, i.e., all-sky effect stronger than clear-sky, semidirect effect negative).

p. 7, l. 21: remove "possible"

p. 8, l. 3: it might be good to add a little more discussion of these results in light of the prevailing opinion that preindustrial/background aerosol properties constitute a large (and irreducible) uncertainty on the anthropogenic forcing; perhaps this result suggests that this problem is not so severe?

p. 8, l. 7: "to" → "from"

p. 8, l. 12: In light of this statement, it would be interesting to see the clear-sky values added to Tab. 2. If the Twomey and direct effects are relatively close in MAC–SP, then it would not be surprising that cloud parameterization doesn't affect the forcing.

Sec. 3.5: I like this additional text very much. I believe you should rephrase the sentence starting on p. 12, l. 7 ("A clear saturation...") to reflect more clearly that the largest efficiencies occur at the (arbitrarily chosen) edges of the plumes, one of the reasons we would expect MAC–SP to provide a lower bound on the strength of the Twomey effect. (See my first comment on the previous manuscript version.)

p. 13, l. 19: I thank the authors for humoring my comment on nudging (which I realize in retrospect made no sense in the context of their methods). I think it is good to include this conclusion, and I agree with it except for the part about interfering with adjustments. For example, Ghan et al. (2016), of which some of this study's authors are also coauthors, derive adjustments based on nudged simulations.

Figure 4 b: update the label to "interannual"

---

## Author Response (AR2)

**Response to the comments of reviewer #3**

Thank you for reviewing our revised manuscript and your recommendation for further improving the text. Please find our point-by-point reply in blue underneath your comments in black.

I have reviewed the revised version of Fiedler et al. Since I was not part of the first round of review, I have mostly restricted myself to determining whether the authors have addressed the comments raised by the other reviewers. However, I am a bit disappointed that those reviewers did not bring up the most poignant criticism of the "simple plumes" (SP) parameterization of aerosol–cloud interactions (ACI), which is that the sporadic transport of anthropogenic pollution into usually very clean regions is underrepresented.

MACv2-SP represents the monthly mean distribution of anthropogenic aerosol optical properties and the associated Twomey effect. The mean transport into relatively clean ocean regions is therefore represented by MACv2-SP, but sub-monthly variability of anthropogenic aerosol is not simulated. The parameterization is deliberately kept simple, e.g., to enable easy experimentation and to ensure a computationally cheap representation of anthropogenic aerosol (Stevens et al., 2017). In the revised section 2.1, we note: „ By design, MACv2-SP does not simulate sub-monthly variability in anthropogenic aerosol. (…) Stevens et al. (2017) give further details on MACv2-SP.", and „Note that $\eta_N$ is only available for regions with $\tau_a > 0$ (see Fig.1)"

My biggest concern with the manuscript is one raised by both of the original reviewers, which is that the manuscript is difficult to read. This is in part because vague phrasing abounds. One of the reviewers explicitly asked the authors to go through the text line by line to rephrase and reduce potential for confusion. That the authors have made only cursory improvements in the revision is disrespectful of the reviewers (and the manuscript's future readers). It also made me feel antagonistic enough that I seriously considered recommending rejection before deciding on major revisions; after all, what assurance do I have that the authors will consider my recommendations any more than the original reviewers'?

We have substantially changed the text in the first revision in response to the first two reviewers and regret that the resulting language is not as easily accessible as we had hoped for. We welcome the opportunity to further sharpen the text and focus on making our study comprehensible for a larger readership. To do so, we revised the text line by line. We hope this process improved the clarity of the language such that also readers with a background different from our own can more easily access the content of the article. Should there still be any phrasing that you perceive as unclear, do not hesitate to tell us. Please let us add that we are grateful for every constructive criticism and value the recommendations of all our reviewers that help improving the manuscript.

**Main comments**
**1 Major organizational issues**
That being said, I also disagree with multiple comments made by the original reviewer who recommended rejection. While the manuscript currently reads like a grab bag of resultlets, none of which is fully developed into an interesting conclusion (as the original reviews pointed out), a good-faith effort at reorganizing the content, as well as rewriting to remove platitudes and vagueness, would lead to a manuscript well worth reading. I have collected a few suggestions for results to emphasize and discuss in greater depth: On the subject of the averaging required for reliable forcing estimates, have the authors considered the "nudging" approach, where the large-scale

dynamics of the model is constrained to reanalysis (e.g., Zhang et al., 2014, ACP)? Can nudging shorten the integration periods or reduce the averaging required to make reliable estimates of transient forcing time evolution? What are the drawbacks or trade-offs that need to be considered?

Nudging is not a suitable approach for our research question, since we aim to quantify ERF differences for the free-running simulations of the participating climate models. We are interested in determining the ERF of these models, not an ERF resulting from a mixture of these models with re-analysis that nudging would cause. In nudged simulations, the atmospheric fields of a climate model are adjusted to re-analysis data, typically every six hours. Due to the frequent constraint on the model from re-analysis, the model's own rapid adjustments must not necessarily develop like they do in a free-running simulation. As such, we obtain an ERF different from the one of the free-running simulations. The benefit of nudging is constraining the synoptic-scale circulation in the simulation, e.g., Zhang et al. (2014), but it is not clear whether nudging shortens the time period required for estimating the model's ERF, since also re-analysis represents natural variability. We add in Section 3.4: „The result underlines again the importance of using a large number of simulated years for determining changes in ERF from free-running climate models. ", and explicitly comment on nudging in the conclusions: „The interannual variability in ERF, and hence the number of years needed to estimate ERF, could be different in nudged model simulations (e.g., Zhang et al., 2014). However, nudging a model simulation with re-analysis data can change the climatology and interfere with the rapid adjustments. The resulting ERFs from a nudged simulation are therefore likely different compared with free-running model simulations."

The point that it may be more fruitful to calculate forcing differences between different levels of anthropogenic pollution rather than a forcing relative to a poorly characterized "preindustrial" state is well taken. I should note that I do not understand why this point appears in the section it appears in, "Benchmarking RF" (but I also do not understand what "benchmarking" means as used by the authors).

We change the organization of this section and now split the content into two subsections. The above mentioned point now appears in the second subsection „3.4 Uncertainties in RF" following the subsection with the title „3.3 Contributions from RF and adjustments". We also rephrase „benchmarking" in the entire text, e.g.: „The offline radiation-transfer model is used to assess the role of uncertainty in (…) " Please refer to the manuscript with tracked changes for reviewing further changes.

I believe the original reviewer's comment about the spatial shift in pollution from Europe/North America to Asia is wrong;(Parenthetically, I also think the reviewer comment about ensemble means is incorrect; the sample mean is the optimal estimate of the population mean regardless of the caveats the reviewer lists. There is absolutely no guarantee that it will converge on the true forcing, but that is an inherent problem in modeling and unrelated to the number of models that participate.) contrary to that reviewer's opinion, the fact that the AOD is similar in both time periods does not imply that the ERF or RF should necessarily be similar. However, for this finding to be illuminating, the authors should endeavor to explain why this is the case rather than simply stating the fact in Sec. 3.4.

We agree that the same global mean AOD does not imply that the forcing is the same. We hope the additional work on the text makes that point even clearer for our readers, e.g., we add in Section 3.5 (former Section 3.4): „Although the global mean $\tau_a$ is similar for 1975 and 2005, the anthropogenic pollution covers very different regions, with the largest maxima in Europe and the U.S.

during the mid-1970s and in East Asia during the mid-2000s. The regional differences in clouds, insolation and surface albedo can contribute to changes in the radiative effects that can result in a different global ERF. For instance, Figure A1-A3 show the spatial differences for cloud properties and the surface albedo illustrating both the regional differences and the model diversity for their representation (see Appendix B). (…)"

The revised manuscript additionally shows the surface albedo for shortwave radiation from our model ensemble in Appendix B3: „An additional influence on the radiative forcing of anthropogenic aerosol is the surface reflectivity for shortwave radiation. We therefore document the surface albedo for shortwave radiation from the participating models and the satellite product used in the offline radiative transfer calculations of this study. In the global mean, the models and the satellite product are very similar, with a surface albedo of 14-16\%. However, the spatial distributions in Figure A3 indicate differences. The typical difference between less reflective ocean surfaces compared to land regions is apparent. Moreover, the analysis reveals diversity in the regional surface albedos of the participating models, typically related to areas affected by snow cover. Since such diversity in the surface albedo was already previously reported for aerosol-climate models with implications for the aerosol radiative forcing (e.g., Stier et al., 2007), future efforts are still needed for constraining the surface albedo in complex models.".

Moreover, we introduce a more detailed analysis of the forcing efficiency for explaining the reasons for the same global ERF from the substantial different patterns. These results are shown in Section 3.5 (former section 3.4). Please refer to our reply to the next comment for these changes.

The same holds for the discussion of the differences in efficacy. Why is the efficacy different between the time periods?
In the revised manuscript, we investigate the clear- and cloudy-sky contributions to the all-sky efficiency for better illustrating the reasons for the regional differences in the efficiency resulting in the same global ERF. Please note that we also replaced „efficacy" with „efficiency" for a clearer distinction to the efficacy of forcing agents for temperature responses. In the revised manuscript, we replace Tables 3 and 4 with the new Figure 9 that shows the spatial distribution of the efficiency for present-day and the change relative to the mid-1970s for the all-, clear- and cloudy sky.

We change in Section 3.5 (former 3.4): „ The cloudy- and clear-sky contributions to the all-sky efficiency of the ERF, in other words the ratio of ERF to $\tau_a$ helps to better understand why the two $\tau_a$ patterns yield similar ERFs. All-sky efficiency is the sum of contributions from cloudy and clear-sky:

$$\frac{\mathrm{ERF}_{\mathrm{all}}}{\tau_a} = f \frac{\mathrm{ERF}_{\mathrm{cloudy}}}{\tau_a} + (1-f)\frac{\mathrm{ERF}_{\mathrm{clear}}}{\tau_a},$$

where f is the total cloud fraction, and $\mathrm{ERF}_{\mathrm{cloudy}}$ and $\mathrm{ERF}_{\mathrm{clear}}$ the ERF in cloudy and clear sky, respectively.

Figure 9 shows the regional distribution from the multi-model ensemble average of the terms of Equation 1. The all-sky efficiency often increases with increasing distance to major pollution sources because of the decreasing background aerosol, up to -100$Wm^{-2}$ per unit $\tau_a$. These allsky efficiencies are primarily explained by the cloudy-sky contributions. A clear saturation of aerosol-cloud interactions towards the edges of the $\tau_a$ plumes is not evident and the spatial distribution of the all- and cloudy-sky efficiency is rather inhomogeneous. The inhomogeneity contrasts with the clear-sky efficiency, which has much smaller spatial variability and changes only weakly with the $\tau_a$ patterns of the mid-2000s and the mid-1970s.

Averaged globally, all-sky forcing efficiencies for the two aerosol patterns are similar at -26 $Wm^{-2}$ per unit $\tau_a$. The regional all-sky ERF efficiencies, however, change between the mid-1970s and mid-2000s (Fig. 9). This change is almost exclusively explained by the cloudy-sky contribution to the ERF efficiency, reflecting the regional change in $\eta_N$ from the mid-1970s to mid-2000s. The strong change in the cloudy-sky contribution is in strong contrast to the relatively minor changes in the clear-sky contributions. Differences in regional efficiencies of anthropogenic aerosol effects on clouds thus balance in the global mean and result in similar global ERFs for the mid-1970s and mid-2000s.

Of all models, NorESM and EC-Earth have the strongest effective radiative forcing efficiencies around -30 and -40 $Wm^{-2}$ per unit $\tau_a$, respectively, i.e., the same aerosol perturbation in these two models is much more efficient in inducing effective radiative effects than in the other models, consistent with the more negative ERFs (Fig. 8). In EC-Earth, the more negative ERF arises from also perturbing the cloud microphysics with $\eta_N$. In NorESM, the more negative ERF arises from a strong negative RF and a small net contribution from adjustments."

(As a side note, the numbers on p. 8 l. 26 are slightly different from the original manuscript; why is that?)
The ensemble-mean ERF values in the second version of the paper differ slightly from the first version, because EC-Earth was not yet included in the first version.

Why do the efficacies increase in the less polluted regions? I would assume this is because the ACI sensitivity saturates. If so, what does that imply for the reliability of the SP method, where the model sees essentially the same average concentration of anthropogenic pollution, while in model configuration where the model is allowed to do its own transport, anthropogenic aerosol can sporadically intrude into clean, and therefore highly sensitive, regions? (I would say it indicates that the SP method will lead to a significant underestimate of the RFaci, but my point here is that the authors need to discuss their findings.)
We include a new assessment of the clear, cloud and all-sky efficiency of the radiative effects of anthropogenic aerosol from our model ensemble. It shows that the efficiency often increases in less polluted regions because the aerosol optical depth in the denominator of $\eta_N$ is smaller than closer to pollution sources. However, the new spatial assessment of the cloudy-sky efficiency of the radiative effect also illustrates that this is not as spatially homogenous like the aerosol plumes would suggest. Away from the centre of the plumes, the relatively smaller anthropogenic AOD has a relatively larger radiative effect that one would also expect with sub-monthly variability in aerosol transport. We include the new Figure 9 and have changed the text in the section, e.g.: „These all-sky efficiencies are primarily explained by the cloudy-sky contributions. A clear saturation of aerosol-cloud interactions towards the edges of the $\tau_a$ plumes is not evident and the spatial distribution of the all- and cloudy-sky efficiency is rather inhomogeneous. The inhomogeneity contrasts with the clear-sky efficiency, which has much smaller spatial variability and changes only

weakly with the $\tau_a$ patterns of the mid-2000s and the mid-1970s.", and we also state explicitly that MACv2-SP does not simulate sub-monthly aerosol variability in Section 2.1.

Continuing on the previous point, in the introduction, the authors say that one of their research questions is how differences in surface albedo, insolation, and cloud regimes affect ERF over time. However, they do not return to this question in the manuscript. If they follow my suggestion in my previous point, it will have the cobenefit of making their introduction more reflective of the paper. We think this comment is based on the discussion article that we already have revised. Maybe the outdated version of the manuscript has also unintentionally been used for other parts of the present review which could explain the perception that we have not done enough in response to the first reviews. The phrasing of the research questions in the already revised introduction is: (…) „Does the substantial spatial change of the anthropogenic aerosol between the mid-1970s and mid-2000s affect the global magnitude of ERF?" using ensembles of simulations from five global aerosol-climate models with reduced aerosol complexity. In this context, we additionally ask: "What is the relative contribution of variability amongst and within models to the spread in ERF?"

With the additional work on the text, the part with the research questions now is: „(…) „Does the substantial spatial change of the anthropogenic aerosol between the mid-1970s and mid-2000s affect the global magnitude of ERF?", based on ensembles of simulations from five global aerosol-climate models, all using identical anthropogenic aerosol perturbations of reduced complexity. In this context, we additionally ask: "What is the relative contribution of internal model variability to the ERF spread?", and document the model diversity for the pre-industrial aerosol as well as cloud characteristics and the surface albedo that are relevant for the ERF of anthropogenic aerosol. "

We therefore list potentially different quantities that affect radiative effects for the two patterns and revisit them in Section 3.5. Please refer to the text changes in the manuscript or aloft. Additionally, the revised manuscript has now a new section on the surface albedo (Appendix B3). Please also refer to our replies above.

**2 Clarity of writing**
Regarding clarity of the writing, one of the imprecisions that was a constant irritant was the definition of Faci: I think it might mean ERFaci for EC-Earth and RFaci for the other models, but I still haven't been able to figure it out for sure. As for Fari, I'm pretty sure that is the ERFari, from the description on page 4. But if Fx refers to ERF for x = ari and RF for x = aci, that is extremely confusing.
We introduce the abbreviations in the text where they are used for the first time, but we largely remove short forms in the abstract and conclusions for making them clear without the need to read the entire manuscript.

$F_{aci}$ stands for aerosol-cloud interaction (defined in Section 2.1), RF for instantaneous radiative forcing and ERF for effective radiative forcing (defined at the beginning).

We explain the implementation of $F_{aci}$ in our models in Section 2.1:"MACv2-SP mimics the spatio-temporal distribution and wavelength dependence of the optical properties of anthropogenic aerosols as well as a change in the cloud droplet number concentration (N) to represent aerosol-radiation interactions ($F_{ari}$) and aerosol-cloud interactions ($F_{aci}$) in a consistent manner. (…) All

models account for the first indirect or Twomey effect by multiplying their cloud droplet number concentrations, calculated for pre-industrial aerosol conditions, by $\eta_N$ prior to the radiative transfer calculation. Since $\eta_N$ is larger than one in the presence of anthropogenic aerosols, the effective radius of cloud droplets is reduced, which enhances the cloud reflectivity of shortwave radiation. In addition, the EC-Earth model also includes a second indirect or cloud lifetime effect by using the modified cloud droplet number concentrations in the cloud microphysics scheme (Döscher et al., 2018)."

The original reviewers went to some trouble to identify other particularly unclear passages. I am somewhat deterred by the lack of response by the authors, so I will not expend effort on listing further instances. Just by way of example, in the first paragraph of the conclusions: What does "the" in "the five state of the art models" mean? It makes it sound like this is an exhaustive list, so no other models are state of the art. I know that this is not what the authors mean, but the sloppy writing is doing them a disservice.

We have changed the first sentences of the conclusion to: „We assess the radiative effects of anthropogenic aerosol in ensembles of simulations from five state-of-the-art aerosol climate models, prescribing identical anthropogenic aerosol properties of reduced complexity. Each of the participating models uses annually repeating patterns of anthropogenic aerosol for obtaining 180 years of radiative forcing estimates. The multi-model multi-ensemble present-day all-sky short-wave effective radiative forcing (…)". Additionally to the already changed language after the first revision, we also went through the manuscript line-by-line again and hope that the other editorial changes make the language accessible for a larger audience.

"reflecting both natural variability and model differences affecting ERF" is such an unclear way of restating the previous clause in that sentence that it took me forever to figure out that is was meant as a restatement. Writing something like "reflecting that natural variability and model differences both contribute to the model diversity in ERF" would have made it clear immediately. I know that it is hard to identify unclear passages in something that one has written oneself, but this paper has 12 authors, so there was no shortage of opportunity for someone to approach the text in the role of an uninitiated reader.

We have changed this section. Please let us add that we have collaborated on the writing of the manuscript and regret that the resulting writing style is not as easy to understand as we had hoped for. We are grateful for your open words and gladly follow your suggestion of using the manpower of this article to further improve the comprehension of our text.

What are the "best" model-mean estimates? (For that matter, what does "model-mean" mean in that sentence?)

Changed to: „(…) we obtain an ERF spread of -0.9 to -0.4 $Wm^{-2}$ associated with systematic model differences" in the conclusions".

I do implore the authors to follow the suggestion of having someone (who will receive better compensation than a reviewer) go through the manuscript line by line to improve the writing.

We edited the text line by line with a focus on making the content easier to understand for a diverse readership and hope the revised language prevents confusion in the future. The way we approached the language changes this time was as follows. Firstly, three different co-authors revised the entire manuscript line by line. Secondly, all co-authors had the opportunity to review

and comment on the revised manuscript. Finally, two authors did quick reading of the manuscript for determining whether the text clearly conveys the message. We hope this process helped making the content of the article easier accessible for a large readership with a diverse background.

**3 Minor comments**

p. 5, l. 32 onward: how can you tell this is not just coincidence?

We perform experiments for obtaining 180 estimates of ERF for each model. Using the long-term averaged ERFs of the models gives us confidence that the results are not just obtained by coincidence. We change: „The long-term averaged ERFs of ECHAM and ECHAM-HAM are similar, despite ECHAM using a prescribed climatology of $\tau_p$ and ECHAM-HAM simulating $\tau_p$ interactively (Section 2.1). This similarity suggests that the sub-monthly variability of natural aerosol does not substantially affect the mean ERF of anthropogenic aerosol, as long as $F_{\mathrm{aci}}$ is treated consistently in the two models."

p. 6, l. 28: I do not understand this sentence; what does "more than one model ensemble" refer to? More than an ensemble of runs from one model? More than the multi-model ensemble in this study?

Changed to: „Taken together, the size of year-to-year variability and regional model differences in contributions to the global ERF imply that an ensemble of simulations with more than one model, as done here, is needed for constraining the radiative effect of anthropogenic aerosol regionally."

p. 7, l. 1: "are" ! "is"

Replaced.

The word "herein" appears frequently, and I don't think I understood what it was supposed to mean once

Removed throughout.

I like the appendices, and I do not understand why the original reviewer complained about "too much detail" in the model description (the "too many notes, Mr. Mozart" line from Amadeus comes to mind). I believe the long form of "DMS" is two words (p. 10, l. 29). While "vertically integrated liquid water content" (p. 12, l. 25) is correct, why not call it by its better known name, liquid water path?

Changed to: „dimethyl sulphide" and „liquid water path".

In the author contributions, "lead" ! "led"

Replaced.

In Tab. 3, what does it mean when a clear-sky ERF is more negative than a cloudy-sky ERF? Positive forcing by ACI?

The all-sky ERF is smaller than the clear-sky value because of the masking by clouds. We list these values in Table 2 and discuss it in the second paragraph of Section 3.1 where we have worked on the language as in the rest of the manuscript: „The all-sky ERFs from the models are 10-50% smaller than the clear-sky ERF in all models, except in EC-Earth, because clouds mask

the ERF of low-level anthropogenic aerosol (Table 2). That masking by clouds is most pronounced in HadGEM3. In EC-Earth, the all-sky ERF is more negative than in clear-sky because EC-Earth includes cloud lifetime effects of anthropogenic aerosols, thus simulating a stronger $F_{\mathrm{aci}}$."

Please note an additional change during this revision. We decided to not show the instantaneous radiative forcing (RF) from the double radiation calls in EC-Earth to avoid confusion, because the double radiation calls in EC-Earth do not account for the Twomey effect like it was done for the RF estimates in the other models.

[revised manuscript text omitted]

---

## Author Response (AR3)

**Response to the comments of reviewer #3**

I commend the authors on a much improved manuscript. The significant investment of effort into a sentence-by-sentence rewriting of the manuscript by the entire author list has made the conclusions much easier for the reader to grasp. As a minor comment, since "larger audience" is mentioned multiple times in the authors' reply, I do want to clarify that the issue was never that the reader had to be an expert, but rather that even the experts were left confused.

My very minor comments use the differences document attached to the authors' responses because this was the only document I had access to at the time due to computer problems while traveling.

We thank the reviewer for the appraisal. We are glad that our effort in revising the language helped making the text clearer. In the following, we reply (blue) to your comments (black).

p. 4, l. 10: I am still unclear on the definition of Fari (and aci). From the author replies, I am led to believe these are not radiative flux differences, but rather . . . what? The physical processes themselves?

We changed it to: „(…) to induce radiative effects associated with the physical processes of aerosol-radiation interactions ($F_{ari}$) and aerosol-cloud interactions ($F_{aci}$) in a consistent manner."

p. 5, l. 25: what does "interpretation" mean?

Changed to: „MACv2 is also based on more recent emission data relative to 1850 (…)"

p. 6, l. 18: I know what you mean, but you might want to rewrite this sentence to make it easier to understand why this is efficient;

We add the reason:„ The six pre-industrial simulations serve as the reference for the experiments with anthropogenic aerosol and therefore efficiently increase the number of forcing estimates for anthropogenic aerosol."

on the previous line, "three experiment" ! "three experiments"

Changed as suggested.

p. 7, l. 13: state explicitly that "ADJ" is the semidirect effect except for EC-EARTH.

Added: „Our rapid adjustments are associated with atmospheric temperature changes, i.e., semi-direct effects, except for EC-Earth accounting also for adjustments in cloud microphysics."

I am a little confused that the adjustments are positive; I thought that they were typically negative in GCMs. Is this because the Twomey effect is weaker than the direct aerosol effect in MAC–SP (which I think is also unusual)? It might be good to provide a few sentences of context for these MAC–SP results compared to other multi-model ensembles (AeroCom, CMIP5); I think most readers will be convinced that they should expect lower intermodel spread when the Twomey effect is prescribed, but they may share my surprise when effects change sign compared to the conventional wisdom, i.e., all-sky effect stronger than clear-sky, semidirect effect negative).

When MACv2-SP is implemented in the radiation calculation, the net contribution from rapid adjustment is positive. This is the case for all our models except EC-Earth and consistent with buffering of perturbations in the atmosphere. Most GCMs, however, perturb also the cloud microphysics. This is expected for models implementing MACv2-SP in the cloud microphysics like

EC-Earth that induces also rapid adjustments in cloud processes. Such negative adjustments are consistent with your perception of GCM results. These can be large in models, but their magnitudes remain uncertain (Bellouin et al., in prep.).

The Twomey effect from the standard setup of MACv2-SP in ECHAM is about the same size as the direct radiative forcing, namely -0.27Wm$^{-2}$ and -0.23Wm$^{-2}$ (Fiedler et al., 2017). This relative contribution is in agreement with the last assessment report by the IPCC (Myhre et al., 2013). It has been previously discussed for MACv2-SP in Fiedler et al. (2017) alongside an assessment of the impact of the parameterization of the Twomey effect and a discussion of a more negative present-day forcing of -1Wm$^{-2}$ when a stronger Twomey effect is induced with MACv2-SP. The magnitude of the Twomey effect is therefore easy to adapt with MACv2-SP, when the magnitude will be better constrained, but it currently remains uncertain (Bellouin et al., in prep.).

We added in Section 3.3: „The positive net contribution from adjustments is consistent with buffering of perturbations by atmospheric processes. A discussion of the rapid adjustments and the choice for the Twomey effect in ECHAM is given by Fiedler et al. (2017)." The references to previous inter-comparisons from Myhre et al. (2013) and Shindell et al. (2013) are added in Section 3.1: „One could expect less model diversity in all-sky ERF from our study than from previous inter-comparison projects (e.g., Myhre et al., 2013, Shindell et al., 2013), because we prescribe the same aerosol optical properties and the associated change in cloud droplet numbers."

p. 7, l. 21: remove "possible"
Removed.

p. 8, l. 3: it might be good to add a little more discussion of these results in light of the prevailing opinion that preindustrial/background aerosol properties constitute a large (and irreducible) uncertainty on the anthropogenic forcing; perhaps this result suggests that this problem is not so severe?
We added: „Using different parameterizations for Faci can change this result because of non-linear processes. The magnitude of Faci, however, remains uncertain (Bellouin et al., in prep.). One contributing uncertainty is the poor quantitative understanding of the pre-industrial aerosols (e.g., Carslaw et al., 2013)."

p. 8, l. 7: "to" ! "from"
Changed as suggested.

p. 8, l. 12: In light of this statement, it would be interesting to see the clear-sky values added to Tab. 2. If the Twomey and direct effects are relatively close in MAC–SP, then it would not be surprising that cloud parameterization doesn't affect the forcing.
Table 2 lists the clear-sky ERFs in brackets, illustrating the model diversity in both all- and clear-sky ERF from MACv2-SP. We added: „One could expect less model diversity in all-sky ERF from our study than from previous inter-comparison projects (e.g., Myhre et al., 2013, Shindell et al., 2013), because we prescribe the same aerosol optical properties and the associated change in cloud droplet numbers. However, our model diversity in clear-sky ERF is smaller than for our all-sky ERF (Table 2). This points to the influence of model differences in representing clouds (Appendix B) on the all-sky ERF. Our results therefore indicate that model differences in

meteorological parameters contribute to the model diversity in all-sky ERF. This is also the case for the ERF uncertainty in a complex aerosol-climate model (Regayre et al., 2018)." Please refer to our response aloft for the relative contributions from the Twomey and direct effects in ECHAM.

Sec. 3.5: I like this additional text very much. I believe you should rephrase the sentence starting on p. 12, l. 7 ("A clear saturation. . . ") to reflect more clearly that the largest efficiencies occur at the (arbitrarily chosen) edges of the plumes, one of the reasons we would expect MAC–SP to provide a lower bound on the strength of the Twomey effect. (See my first comment on the previous manuscript version.)

We had added the new figure 9 for illustrating that the largest efficiencies do not necessarily occur at the edges of the plumes. The text is now: „Large efficiencies occur typically in remote areas including some regions at the edges of $\tau_a$ plumes (Fig. 9). No clear saturation of Faci is evident at all edges of the $\tau_a$ plumes. Also the spatial distribution of both the all- and cloudy-sky efficiency is rather inhomogeneous."

The position and extent of the plumes (Stevens et al., 2017) were chosen for reproducing the spatial distributions of the monthly mean in the anthropogenic aerosol optical depth from the MPI-M Aerosol Climatology that has changed between version 1 and version 2 (MAC, Kinne et al., 2013, Kinne, in review, Section 2.1 in the manuscript). Please also note our comment aloft on the uncertainty of the magnitude of the Twomey effect (Bellouin et al., in prep.) as well as the flexibility to induce different strengths and patterns for the Twomey effect with MACv2-SP (Fiedler et al., 2017).

p. 13, l. 19: I thank the authors for humoring my comment on nudging (which I realize in retrospect made no sense in the context of their methods). I think it is good to include this conclusion, and I agree with it except for the part about interfering with adjustments. For example, Ghan et al. (2016), of which some of this study's authors are also coauthors, derive adjustments based on nudged simulations.

There seems to be a debate on the usage of nudging for diagnosing aerosol forcing in the community. Also Ghan et al. (2016) acknowledges that nudging temperatures influences the development of moist convection in some models based on the study by Zhang et al. (2014) and discourages the usage of this type of nudging. We added here: „The interference of nudging with adjustments deserves closer attention in future research."

Figure 4 b: update the label to "interannual"

[revised manuscript text omitted]